# Rotating Features for Object Discovery

**Sindy Löwe**
AMLab
University of Amsterdam

**Phillip Lippe**
QUVA Lab
University of Amsterdam

**Francesco Locatello**
Institute of Science and
Technology Austria (ISTA)

**Max Welling**
AMLab
University of Amsterdam

## Abstract

The binding problem in human cognition, concerning how the brain represents and connects objects within a fixed network of neural connections, remains a subject of intense debate. Most machine learning efforts addressing this issue in an unsupervised setting have focused on slot-based methods, which may be limiting due to their discrete nature and difficulty to express uncertainty. Recently, the Complex AutoEncoder was proposed as an alternative that learns continuous and distributed object-centric representations. However, it is only applicable to simple toy data. In this paper, we present Rotating Features, a generalization of complex-valued features to higher dimensions, and a new evaluation procedure for extracting objects from distributed representations. Additionally, we show the applicability of our approach to pre-trained features. Together, these advancements enable us to scale distributed object-centric representations from simple toy to real-world data. We believe this work advances a new paradigm for addressing the binding problem in machine learning and has the potential to inspire further innovation in the field.

## 1 Introduction

Discovering and reasoning about objects is essential for human perception and cognition [8, 72], allowing us to interact with our environment, reason about it, and adapt to new situations [42, 43, 83]. To represent and connect symbol-like entities such as objects, our brain flexibly and dynamically combines distributed information. However, the binding problem remains heavily debated, questioning how the brain achieves this within a relatively fixed network of neural connections [28].

Most efforts to address the binding problem in machine learning center on slot-based methods [4, 23, 27, 33, 41, 46, 64, 88]. These approaches divide their latent representations into "slots", creating a discrete separation of object representations within a single layer of the architecture at some arbitrary depth. While highly interpretable and easy to use for downstream applications, this simplicity may limit slot-based approaches from representing the full diversity of objects. Their discrete nature makes it challenging to represent uncertainty in the separation of objects; and it remains unclear how slots may be recombined to learn flexible part-whole hierarchies [31]. Finally, it seems unlikely that the brain assigns entirely distinct groups of neurons to each perceived object.

Recently, Löwe et al. [48] proposed the Complex AutoEncoder (CAE), which learns continuous and distributed object-centric representations. Taking inspiration from neuroscience, it uses complex-valued activations to learn to encode feature information in their magnitudes and object affiliation in their phase values. This allows the network to express uncertainty in its object separations and embeds

---

Correspondence to loewe.sindy@gmail.com

37th Conference on Neural Information Processing Systems (NeurIPS 2023).

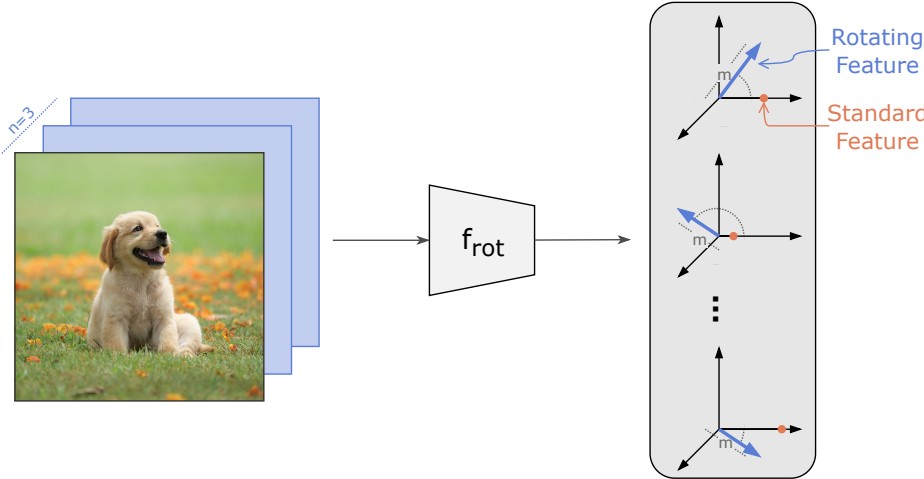

Figure 1: Rotating Features. We propose to extend standard features by an extra dimension $n$ across the entire architecture, including the input (highlighted in blue, here $n = 3$). We then set up the layer structure within $f_{\text{rot}}$ in such a way that the Rotating Features' magnitudes $m$ learn to represent the presence of features, while their orientations learn to represent object affiliation.

object-centric representations in every layer of the architecture. However, due to its two-dimensional (i.e. complex-valued) features, it is severely limited in the number of object representations it can separate in its one-dimensional phase space; and due to the lack of a suitable evaluation procedure, it is only applicable to single-channel inputs. Taken together, this means that the CAE is not scalable, and is only applicable to grayscale toy data containing up to three simple shapes.

In this paper, we present a series of advancements for continuous and distributed object-centric representations that ultimately allow us to scale them from simple toy to real-world data. Our contributions are as follows:

1. We introduce *Rotating Features*, a generalization of complex-valued features to higher dimensions. This allows our model to represent a greater number of objects simultaneously, and requires the development of a new rotation mechanism.

2. We propose a new evaluation procedure for continuous object-centric representations. This allows us to extract discrete object masks from continuous object representations of inputs with more than one channel, such as RGB images.

3. We show the applicability of our approach to features created by a pretrained vision transformer [11]. This enables Rotating Features to extract object-centric representations from real-world images.

We show how each of these improvements allows us to scale distributed object-centric representations to increasingly complicated data, ultimately making them applicable to real-world data. Overall, we are pioneering a new paradigm for a field that has been largely focused on slot-based approaches, and we hope that the resulting richer methodological landscape will spark further innovation.

## 2  Neuroscientific Motivation

Rotating Features draw inspiration from theories in neuroscience that describe how biological neurons might utilize their temporal firing patterns to flexibly and dynamically bind information into coherent percepts, such as objects [15, 20, 21, 50, 53, 65, 79]. In particular, we take inspiration from the temporal correlation hypothesis [66, 67]. It posits that the oscillating firing patterns of neurons (i.e. brain waves) give rise to two message types: the discharge frequency encodes the presence of features, and the relative timing of spikes encode feature binding. When neurons fire in synchrony, their respective features are processed jointly and thus bound together dynamically and flexibly.

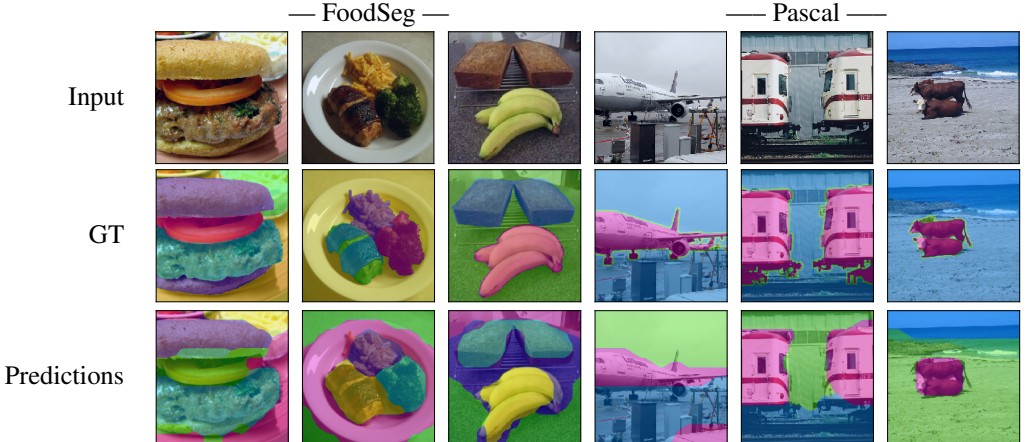

Figure 2: Rotating Features applied to real-world images. By implementing a series of advancements, we scale continuous and distributed object representations from simple toy to real-world datasets.

Previous work in machine learning [48, 58, 59, 60, 61] has taken inspiration from these neuroscientific theories to implement the same two message types using complex-valued features: their magnitudes encode feature presence, and their relative phase differences encode feature binding. However, the one-dimensional orientation information of complex-valued features constrains their expressivity, and existing approaches lack evaluation procedures that scale beyond single-channel inputs. As a result, they are limited to grayscale toy datasets. To overcome these limitations, we propose several advancements for distributed and continuous object-centric representations, which we describe in detail in the next section.

## 3 Rotating Features

We create Rotating Features by augmenting standard features with additional dimensions and by creating a layer structure that parses the magnitude and multi-dimensional orientation of the resulting vectors separately. In the following, we describe how we represent, process, train and evaluate Rotating Features such that their magnitudes learn to represent the presence of features and their orientations learn to represent object affiliation.

### 3.1 Representing Rotating Features

In a standard artificial neural network, each layer produces a $d$-dimensional feature vector $\mathbf{z_{standard}} \in \mathbb{R}^d$, where $d$ might be further subdivided to represent a particular structure of the feature map (e.g. $d = c \times h \times w$, where $c, h, w$ represent the channel, height and width dimensions, respectively). To create Rotating Features, we extend each scalar feature within this $d$-dimensional feature vector into an $n$-dimensional vector, thus creating a feature matrix $\mathbf{z_{rotating}} \in \mathbb{R}^{n \times d}$. Given the network structure described below, its magnitude vector $\left\|\mathbf{z_{rotating}}\right\|_2 \in \mathbb{R}^d$ (where $\|\cdot\|_2$ is the L2-norm over the rotation dimension $n$) behaves similarly to a standard feature vector and learns to represent the presence of certain features in the input. The remaining $n-1$ dimensions represent the orientations of the Rotating Features, which the network uses as a mechanism to bind features: features with similar orientations are processed together, while connections between features with different orientations are suppressed. Note, that we represent Rotating Features in Cartesian coordinates rather than spherical coordinates, as the latter may contain singularities that hinder the model's ability to learn good representations[1].

---

[1]In an $n$-dimensional Euclidean space, the spherical coordinates of a vector $\mathbf{x}$ consist of a radial coordinate $r$ and $n-1$ angular coordinates $\varphi_1, ..., \varphi_{n-1}$ with $\varphi_1, ..., \varphi_{n-2} \in [0, \pi]$ and $\varphi_{n-1} \in [0, 2\pi]$. This representation can lead to singularities due to dependencies between the coordinates. For example, when a vector's radial component (i.e. magnitude) is zero, the angular coordinates can take any value without changing the underlying vector. As our network applies ReLU activations on the magnitudes, this singularity may occur regularly, hindering the network from training effectively.

## 3.2 Processing Rotating Features

We create a layer structure $f_{\text{rot}}$ that processes Rotating Features in such a way that their magnitudes represent the presence of certain features, while their orientations implement a binding mechanism. To achieve this, we largely follow the layer formulation of the CAE [48], but generalize it from complex-valued feature vectors $\mathbf{z}_{\text{complex}} \in \mathbb{C}^d$ to feature matrices of the form $\mathbf{z}_{\text{rotating}} \in \mathbb{R}^{n \times d}$ both in the implementation and in the equations below; and introduce a new mechanism to rotate features. This allows us to process Rotating Features with $n \geq 2$.

**Weights and Biases** Given the input $\mathbf{z}_{\text{in}} \in \mathbb{R}^{n \times d_{\text{in}}}$ to a layer with $n$ rotation dimensions and $d_{\text{in}}$ input feature dimensions, we apply a neural network layer parameterized with weights $\mathbf{w} \in \mathbb{R}^{d_{\text{in}} \times d_{\text{out}}}$ and biases $\mathbf{b} \in \mathbb{R}^{n \times d_{\text{out}}}$, where $d_{\text{out}}$ is the output feature dimension, in the following way:

$$\boldsymbol{\psi} = f_{\mathbf{w}}(\mathbf{z}_{\text{in}}) + \mathbf{b} \quad \in \mathbb{R}^{n \times d_{\text{out}}} \tag{1}$$

The function $f_{\mathbf{w}}$ may represent different layer types, such as a fully connected or convolutional layer. For the latter, weights may be shared across $d_{\text{in}}$ and $d_{\text{out}}$ appropriately. Regardless of the layer type, the weights are shared across the $n$ rotation dimensions, while the biases are not. By having separate biases, the model learns a different orientation offset for each feature, providing it with the necessary mechanism to learn to rotate features. Notably, in contrast to the CAE, the model is not equivariant with respect to global orientation with this new formulation of the rotation mechanism: when there is a change in the input orientation, the output can change freely. While it is possible to create an equivariant rotation mechanism for $n \geq 2$ by learning or predicting the parameters of a rotation matrix, preliminary experiments have demonstrated that this results in inferior performance.

**Binding Mechanism** We implement a binding mechanism that jointly processes features with similar orientations, while weakening the connections between features with dissimilar orientations. This incentivizes features representing the same object to align, and features representing different objects to take on different orientations.

To implement this binding mechanism, we utilize the same weights as in Eq. (1) and apply them to the magnitudes of the input features $\|\mathbf{z}_{\text{in}}\|_2 \in \mathbb{R}^{d_{\text{in}}}$:

$$\boldsymbol{\chi} = f_{\mathbf{w}}(\|\mathbf{z}_{\text{in}}\|_2) \quad \in \mathbb{R}^{d_{\text{out}}} \tag{2}$$

We then integrate this with our previous result by calculating the average between $\boldsymbol{\chi}$ and the magnitude of $\boldsymbol{\psi}$:

$$\mathbf{m}_{\text{bind}} = 0.5 \cdot \|\boldsymbol{\psi}\|_2 + 0.5 \cdot \boldsymbol{\chi} \quad \in \mathbb{R}^{d_{\text{out}}} \tag{3}$$

This binding mechanism results in features with similar orientations being processed together, while connections between features with dissimilar orientations are suppressed. Given a group of features of the same orientation, features of opposite orientations are masked out, and misaligned features are gradually scaled down. This effect has been shown by Löwe et al. [48], Reichert & Serre [61], and we illustrate it in Fig. 3. In contrast, without the binding mechanism, the orientation of features does not influence their processing, the model has no incentive to leverage the additional rotation dimensions, and the model fails to learn object-centric representations (see Appendix D.3). Overall, the binding mechanism allows the network to create streams of information that it can process separately, which naturally leads to the emergence of object-centric representations.

**Activation Function** In the final step, we apply an activation function to the output of the binding mecha-

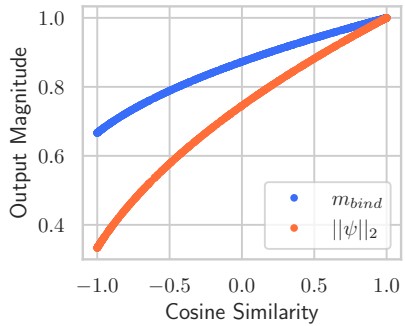

Figure 3: Effect of the binding mechanism. We start by randomly sampling two column vectors $\mathbf{a}, \mathbf{b} \in \mathbb{R}^n$ with $\|\mathbf{a}\|_2 = \|\mathbf{b}\|_2 = 1$. Assuming $d_{\text{in}} = 3, d_{\text{out}} = 1$ and $f_{\mathbf{w}}$ is a linear layer, we set $\mathbf{z}_{\text{in}} = [\mathbf{a}, \mathbf{a}, \mathbf{b}]$, weights $\mathbf{w} = \left[\frac{1}{3}, \frac{1}{3}, \frac{1}{3}\right]^T$ and biases $\mathbf{b} = [0, ..., 0]^T$. Subsequently, we plot the cosine similarity between $\mathbf{a}$ and $\mathbf{b}$ on the x-axis, and $\mathbf{m}_{\text{bind}}$ and $\|\boldsymbol{\psi}\|_2$ on the y-axis, representing the magnitudes of the layer's output before applying the activation function with (blue) and without (orange) the binding mechanism. Without the binding mechanism, misaligned features are effectively subtracted from the aligned features, resulting in smaller output magnitudes for $\|\boldsymbol{\psi}\|_2$. The binding mechanism reduces this effect, leading to consistently larger magnitudes in $\mathbf{m}_{\text{bind}}$. In the most extreme scenario, features with opposite orientations (i.e., with a cosine similarity of -1) are masked out by the binding mechanism, as the output magnitude ($\frac{2}{3}$) would be the same if $\mathbf{z}_{\text{in}} = [\mathbf{a}, \mathbf{a}, \mathbf{0}]$.

nism $\mathbf{m}_{\text{bind}}$, ensuring it yields a positive-valued result. By rescaling the magnitude of $\psi$ to the resulting value and leaving its orientation unchanged, we obtain the output of the layer $\mathbf{z}_{\text{out}} \in \mathbb{R}^{n \times d_{\text{out}}}$:

$$\mathbf{m}_{\text{out}} = \text{ReLU}(\text{BatchNorm}(\mathbf{m}_{\text{bind}})) \quad \in \mathbb{R}^{d_{\text{out}}} \tag{4}$$

$$\mathbf{z}_{\text{out}} = \frac{\psi}{\|\psi\|_2} \cdot \mathbf{m}_{\text{out}} \quad \in \mathbb{R}^{n \times d_{\text{out}}} \tag{5}$$

## 3.3 Training Rotating Features

We apply Rotating Features to vision datasets which inherently lack the additional $n$ dimensions. This section outlines the pre-processing of input images to generate rotating inputs and the post-processing of the model's rotating output to produce standard predictions, which are used for training the model.

Given a positive, real-valued input image $\mathbf{x}' \in \mathbb{R}^{c \times h \times w}$ with $c$ channels, height $h$, and width $w$, we create the rotating feature input $\mathbf{x} \in \mathbb{R}^{n \times c \times h \times w}$ by incorporating $n - 1$ empty dimensions. We achieve this by setting the first dimension of $\mathbf{x}$ equal to $\mathbf{x}'$, and assigning zeros to the remaining dimensions. Subsequently, we apply a neural network model $f_{\text{model}}$ that adheres to the layer structure of $f_{\text{rot}}$ to this input, generating the output features $\mathbf{z} = f_{\text{model}}(\mathbf{x}) \in \mathbb{R}^{n \times d_{\text{model}}}$.

To train the network using a reconstruction task, we set $d_{\text{model}} = c \times h \times w$ and extract the magnitude $\|\mathbf{z}\|_2 \in \mathbb{R}^{c \times h \times w}$ from the output. We then rescale it using a linear layer $f_{\text{out}}$ with sigmoid activation function. This layer has weights $\mathbf{w}_{\text{out}} \in \mathbb{R}^c$ and biases $\mathbf{b}_{\text{out}} \in \mathbb{R}^c$ that are shared across the spatial dimensions, and applies them separately to each channel $c$:

$$\hat{\mathbf{x}} = f_{\text{out}}(\|\mathbf{z}\|_2) \quad \in \mathbb{R}^{c \times h \times w} \tag{6}$$

Finally, we compute the reconstruction loss $\mathcal{L} = \text{MSE}(\mathbf{x}', \hat{\mathbf{x}}) \in \mathbb{R}$ by comparing the input image $\mathbf{x}'$ to the reconstruction $\hat{\mathbf{x}}$ using the mean squared error (MSE).

## 3.4 Evaluating Object Separation in Rotating Features

While training the network using the Rotating Features' magnitudes, their orientations learn to represent "objectness" in an unsupervised manner. Features that represent the same objects align, while those representing different objects take on distinct orientations. In this section, we outline how to process the continuous orientation values of the output Rotating Features $\mathbf{z} \in \mathbb{R}^{n \times c \times h \times w}$, in order to generate a discrete object segmentation mask. This approach follows similar steps to those described for the CAE [48], but introduces an additional procedure that enables the evaluation of Rotating Features when applied to inputs with $c \geq 1$.

As the first step, we normalize the output features such that their magnitudes equal one, mapping them onto the unit (hyper-)sphere. This ensures that the object separation of features is assessed based on their orientation, rather than their feature values (i.e. their magnitudes):

$$\mathbf{z}_{\text{norm}} = \frac{\mathbf{z}}{\|\mathbf{z}\|_2} \quad \in \mathbb{R}^{n \times c \times h \times w} \tag{7}$$

Subsequently, we mask features with small magnitudes, as they tend to exhibit increasingly random orientations, in a manner that avoids trivial solutions that may emerge on feature maps with $c > 1$. For instance, consider an image containing a red and a blue object. The reconstruction $\mathbf{z}$ would be biased towards assigning small magnitudes to the color channels inactive for the respective objects. If we simply masked out features with small magnitudes by setting them to zero, we would separate the objects based on their underlying color values, rather than their assigned orientations.

To avoid such trivial solutions, we propose to take a weighted average of the orientations across channels, using the thresholded magnitudes of $\mathbf{z}$ as weights:

$$\mathbf{w}_{\text{eval}}^{i,j,l} = \begin{cases} 1 & \text{if } \|\mathbf{z}\|_2^{i,j,l} > t \\ 0 & \text{otherwise} \end{cases} \tag{8}$$

$$\mathbf{z}_{\text{eval}} = \frac{\sum_{i=1}^c \mathbf{w}_{\text{eval}}^i \circ \mathbf{z}_{\text{norm}}^i}{\sum_{i=1}^c \mathbf{w}_{\text{eval}}^i + \varepsilon} \quad \in \mathbb{R}^{n \times h \times w} \tag{9}$$

where we create the weights $\mathbf{w}_{\text{eval}}$ by thresholding the magnitudes $\|\mathbf{z}\|_2$ for each channel $i \in [1, ..., c]$ and spatial location $j \in [1, ..., h], l \in [1, ..., w]$ using the threshold $t$ and then compute the weighted

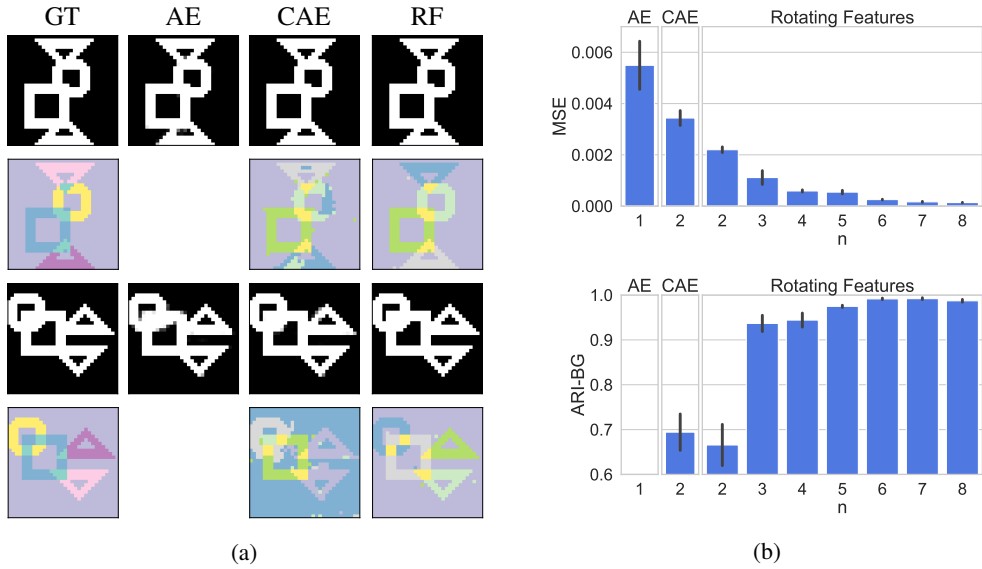

Figure 4: Qualitative and quantitative performance comparison on the 4Shapes dataset. **(a)** In comparison to a standard autoencoder (AE) and the Complex AutoEncoder (CAE), Rotating Features (RF) create sharper reconstructions (1st and 3rd row). Additionally, Rotating Features separate all four shapes, while the CAE does not (2nd and 4th row). **(b)** Larger rotation dimensions $n$ lead to better reconstruction performance (top) and better object discovery performance (bottom), with $n \geq 6$ resulting in a perfect separation of all shapes (mean $\pm$ sem performance across four seeds).

average across channels $c$. Here, $\circ$ denotes the Hadamard product, $\varepsilon$ is a small numerical constant to avoid division by zero, and $\mathbf{w}_{\text{eval}}$ is repeated across the $n$ rotation dimensions.

Lastly, we apply a clustering procedure to $\mathbf{z}_{\text{eval}}$ and interpret the resulting discrete cluster assignment as the predicted object assignment for each feature location. Our experiments demonstrate that both $k$-means and agglomerative clustering achieve strong results, thus eliminating the need to specify the number of objects in advance.

## 4  Experiments

In this section, we evaluate whether our proposed improvements enable distributed object-centric representations to scale from simple toy to real-world data. We begin by outlining the general settings common to all experiments, and then proceed to apply each proposed improvement to a distinct setting. This approach allows us to isolate the impact of each enhancement. Finally, we will highlight some advantageous properties of Rotating Features. Our code is publicly available at github.com/loeweX/RotatingFeatures.

**General Setup**  We implement Rotating Features within a convolutional autoencoding architecture. Each model is trained with a batch-size of 64 for 10,000 to 100,000 steps, depending on the dataset, using the Adam optimizer [40]. Our experiments are implemented in PyTorch [54] and run on a single Nvidia GTX 1080Ti. See Appendix C for more details on our experimental setup.

**Evaluation Metrics**  We utilize a variety of metrics to gauge the performance of Rotating Features and draw comparisons with baseline methods. Reconstruction performance is quantified using mean squared error (MSE). To evaluate object discovery performance, we employ Adjusted Rand Index (ARI) [34, 57] and mean best overlap (MBO) [56, 64]. ARI measures clustering similarity, where a score of 0 indicates chance level and 1 denotes a perfect match. Following standard practice in the object discovery literature, we exclude background labels when calculating ARI (ARI-BG) and evaluate it on instance-level masks. MBO is computed by assigning the predicted mask with the highest overlap to each ground truth mask, and then averaging the intersection-over-union (IoU) values of the resulting mask pairs. Unlike ARI, MBO takes into account background pixels, thus

measuring the degree to which masks conform to objects. We assess this metric using both instance-level ($MBO_i$) and class-level ($MBO_c$) masks.

## 4.1 Rotating Features Can Represent More Objects

**Setup**  We examine the ability of Rotating Features to represent more objects compared to the CAE [48] using two datasets. First, we employ the 4Shapes dataset, which the CAE was previously shown to fail on. This grayscale dataset contains the same four shapes in each image ($\square, \triangle, \triangledown, \bigcirc$). Second, we create the 10Shapes dataset, featuring ten shapes per image. This dataset is designed to ensure the simplest possible separation between objects, allowing us to test the capacity of Rotating Features to simultaneously represent many objects. To achieve this, we employ a diverse set of object shapes ($\square, \triangle, \bigcirc$ with varying orientations and sizes) and assign each object a unique color and depth value.

**Results**  As depicted in Fig. 4, Rotating Features improve in performance on the 4Shapes dataset as we increase the size of the rotation dimension $n$. For the CAE and Rotating Features with $n = 2$, where the orientation information is one-dimensional, the models struggle to distinguish all four objects. However, as the rotation dimension grows, performance enhances, and with $n = 6$, the model perfectly separates all four objects. Additionally, we observe that the reconstruction performance improves accordingly, and significantly surpasses the same autoencoding architecture using standard features, despite having a comparable number of learnable parameters.

Moreover, Rotating Features are able to differentiate 10 objects simultaneously, as evidenced by their ARI-BG score of $0.959 \pm 0.022$ on the 10Shapes dataset (Fig. 5). In Appendix D.5, we provide a detailed comparison of Rotating Features with $n = 2$ and $n = 10$ applied to images with increasing numbers of objects. Additionally, to determine if Rotating Features can distinguish even more objects, we conduct a theoretical investigation in Appendix D.1. In this analysis, we reveal that as we continue to add points to an $n$-dimensional hypersphere, the minimum cosine similarity achievable between these points remains relatively stable after the addition of the first ten points, provided that $n$ is sufficiently large. This implies that once an algorithm can separate ten points (or objects, in our case) on a hypersphere, it should be scalable to accommodate more points. Consequently, the number of objects to separate is not a critical hyperparameter of our model, in stark contrast to slot-based architectures.

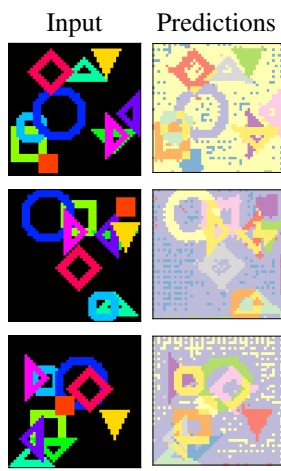

Input    Predictions

Figure 5: Rotating Features learn to separate all ten objects in the 10Shapes dataset.

## 4.2 Rotating Features Are Applicable to Multi-Channel Images

**Setup**  We explore the applicability of Rotating Features and our proposed evaluation method to multi-channel inputs by creating an RGB(-D) version of the 4Shapes dataset. This variation contains the same four shapes per image as the original dataset, but randomly assigns each object a color. We vary the number of potential colors, and create two versions of this dataset: an RGB version and an RGB-D version, in which each object is assigned a fixed depth value.

**Results**  As illustrated in Fig. 6, Rotating Features struggle to distinguish objects in the RGB version of the 4Shapes dataset when the number of potential colors increases. This appears to stem from their tendency to group different shapes together based on shared color. In Appendix D.6, we observe the same problem when applying Rotating Features to commonly used object discovery datasets (multi-dSprites and CLEVR). When depth information is added, this issue is resolved, and Rotating Features successfully separate all objects. This demonstrates that our proposed evaluation procedure makes Rotating Features applicable to multi-channel inputs, but that they require higher-level information in their inputs to reliably separate objects.

## 4.3 Rotating Features Are Applicable to Real-World Images

Inspired by the previous experiment, which demonstrates that Rotating Features and our proposed evaluation procedure can be applied to multi-channel inputs but require higher-level features for optimal object discovery performance, we apply them to the features of a pretrained vision transformer.

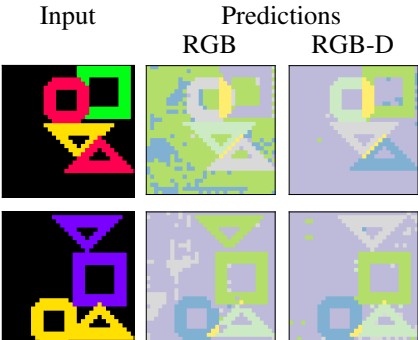
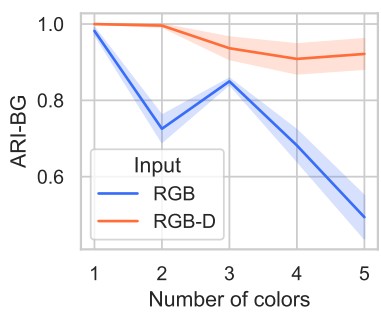

Figure 6: Rotating Features on the 4Shapes RGB(-D) dataset. When applied to RGB images with more colors, object discovery performance in ARI-BG degrades (right; mean $\pm$ sem across 4 seeds) as the Rotating Features tend to group objects of the same color together (left). By adding depth information to the input (RGB-D), this problem can be resolved, and all objects are separated as intended.

**Setup** Following Seitzer et al. [64], we utilize a pretrained DINO model [5] to generate higher-level input features. Then, we apply an autoencoding architecture with Rotating Features to these features. We test our approach on two datasets: the Pascal VOC dataset [19] and FoodSeg103 [85], a benchmark for food image segmentation. On the Pascal VOC dataset, we do not compare ARI-BG scores, as we have found that a simple baseline significantly surpasses previously reported state-of-the-art results on this dataset (see Appendix D.4). On FoodSeg103, we limit the evaluation of our model to the $MBO_c$ score, as it only contains class-level segmentation masks.

**Results** The qualitative results in Fig. 2 highlight the effectiveness of Rotating Features in segmenting objects across both real-world datasets. Table 1 details the object discovery performance of Rotating Features compared to various models on the Pascal VOC dataset. The strongest performance on this dataset is achieved by the *DINOSAUR Transformer* model [64], a Slot Attention-based model with an autoregressive transformer decoder that is applied to DINO pretrained features. Closer to our setting (i.e., without autoregressive models), the *DINOSAUR MLP* model combines Slot Attention with a spatial broadcast decoder (MLP decoder) and achieves the second-highest performance reported in literature in terms of $MBO_c$ and $MBO_i$ scores. The performance of *Rotating Features*, embedded in a comparatively simple convolu-

Table 1: Object discovery performance on the Pascal VOC dataset (mean $\pm$ sem across 5 seeds). Rotating Features' scores indicate a good separation of objects on this real-world dataset. Baseline results adapted from Seitzer et al. [64].

| Model | $MBO_i \uparrow$ | $MBO_c \uparrow$ |
|---|---|---|
| Block Masks | 0.247 | 0.259 |
| Slot Attention | $0.222 \pm 0.008$ | $0.237 \pm 0.008$ |
| SLATE | $0.310 \pm 0.004$ | $0.324 \pm 0.004$ |
| Rotating Features –DINO | $0.282 \pm 0.006$ | $0.320 \pm 0.006$ |
| DINO $k$-means | 0.363 | 0.405 |
| DINO CAE | $0.329 \pm 0.009$ | $0.374 \pm 0.010$ |
| DINOSAUR Transformer | $0.440 \pm 0.008$ | $0.512 \pm 0.008$ |
| DINOSAUR MLP | $0.395 \pm 0.000$ | $0.409 \pm 0.000$ |
| Rotating Features | $0.407 \pm 0.001$ | $0.460 \pm 0.001$ |

tional autoencoding architecture, lies in between these two Slot Attention-based models. Rotating Features also outperform a baseline that applies $k$-means directly to the DINO features (*DINO k-means*), showing that they learn a more meaningful separation of objects. Finally, the performance gain of Rotating Features over the original CAE model applied to the DINO features (*DINO CAE*) highlights their improved object modelling capabilities. For reference, object discovery approaches applied directly to the original input images (*Slot Attention* [46], *SLATE* [68], *Rotating Features –DINO*) struggle to segment the objects in this dataset and achieve performances close to a *Block Masks* baseline that divides images into regular rectangles. To verify that Rotating Features work on more settings, we also apply them to the FoodSeg103 dataset. Here, they achieve an $MBO_c$ of $0.484 \pm 0.002$, significantly outperforming the block masks baseline ($0.296 \pm 0.000$). Overall, this highlights that Rotating Features learn to separate objects in real-world images through unsupervised training; thus scaling continuous and distributed object representations from simple toy to real-world data for the first time.

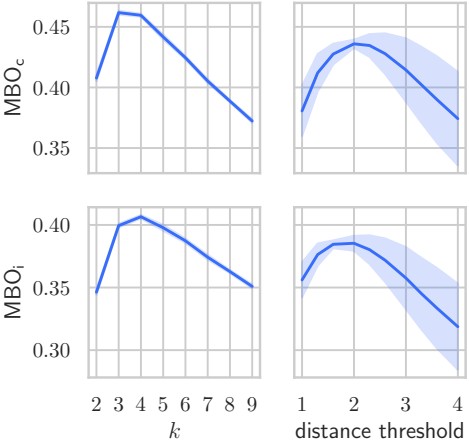
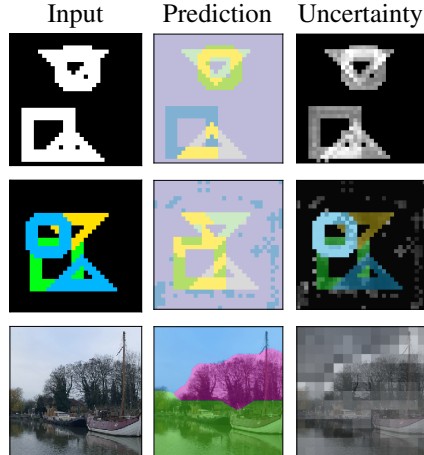

| | Input | Prediction | Uncertainty |

Figure 7: Clustering methods on Pascal (mean $\pm$ sem across 4 seeds). After training, we can efficiently compare various clustering methods and hyperparameters such as $k$-means (left) with different values of $k$ and agglomerative clustering (right) with different distance thresholds.

Figure 8: Uncertainty maps of Rotating Features. We plot the distance of each normalized output feature to its closest k-means cluster center, using brighter pixels for larger distances. This highlights uncertainty in areas of object overlap, incorrect predictions, and object boundaries.

### 4.4 Benefits of Rotating Features

Rotating Features present several advantageous properties: they learn to distinguish the correct number of objects automatically, they generalize well to different numbers of objects than observed during training, they express uncertainty in their object separations, and they are very efficient to run. In this section, we delve into these properties in detail.

**Clustering Methods** We can evaluate various clustering methods and hyperparameters very efficiently on a Rotating Features model, as they can be freely interchanged after training (Fig. 7). Through their continuous object assignments, Rotating Features learn to separate the correct number of objects automatically. By evaluating with agglomerative clustering, the requirement to specify the number of objects to be separated can be eliminated completely, albeit with slightly reduced results.

**Generalization Performance** Rotating Features can generalize beyond the number of objects observed during training, even when the number of objects in the test images is unknown. To highlight this, we train a Rotating Features model and a Slot Attention model [46] on a revised version of the 10Shapes dataset. This version includes the same ten unique shapes, but randomly selects six to be included in each image. After training, we test the models on a range of variants of this dataset, displaying between two and ten objects per image. As shown in Fig. 9, when the number of objects is known and the parameters of the models are set accordingly, both Rotating Features and SlotAttention maintain a relatively stable performance across

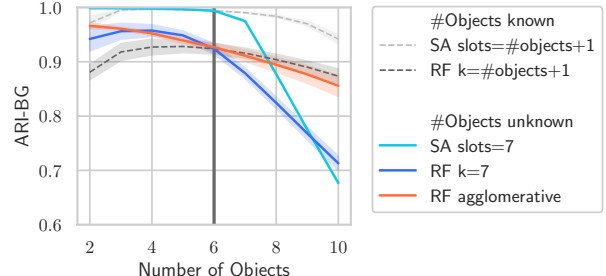

Figure 9: Testing the generalization performance of Rotating Features (RF) and Slot Attention (SA). All models are trained on images containing six objects. When the number of objects in the test images is unknown, only the Rotating Features model evaluated with agglomerative clustering maintains a relatively stable performance across varying numbers of objects.

various numbers of objects per scene; with SlotAttention consistently outperforming the Rotating Features model. However, when the number of objects in the test images is unknown, and we set the parameters of the model according to our best estimate given the training data, the performances of

the SlotAttention model and the Rotating Features model evaluated with $k$-means degrade considerably as the number of objects in the test images increases. This problem can be circumvented by evaluating Rotating Features with agglomerative clustering. After determining the distance threshold on the training dataset, this approach preserves a fairly consistent performance across varying numbers of objects in each scene, considerably outperforming the other approaches that require the number of objects to be set in advance. In summary, our results suggest that Rotating Features can generalize beyond the number of objects observed during training, even when the number of objects in the test images is unknown.

**Uncertainty Maps** Since Rotating Features learn continuous object-centric representations, they can express and process uncertainty in their object separations. We evaluate this by plotting the L2 distance of each normalized output feature to the closest $k$-means center in Fig. 8. This reveals higher uncertainty in areas of object overlap, object boundaries and incorrect predictions.

**Training Time** Rotating Features are very efficient to train. On a single Nvidia GTX 1080Ti, the presented Rotating Features model on the Pascal VOC dataset requires less than 3.5 hours of training. For comparison, the DINOSAUR models [64] were trained across eight Nvidia V100 GPUs.

## 5   Related Work

Numerous approaches attempt to solve the binding problem in machine learning [28], with most focusing on slot-based object-centric representations. Here, the latent representation within a single layer of the architecture is divided into "slots", creating a discrete separation of object representations. The symmetries between slots can be broken in various ways: by enforcing an order on the slots [4, 16, 18], by assigning slots to spatial coordinates [1, 3, 6, 7, 9, 13, 45, 63], by learning specialized slots for different object types [32, 33, 80, 82], by using an iterative procedure [12, 24, 25, 26, 27, 44, 74, 90] or by a combination of these methods [17, 70, 77, 86]. Slot Attention [46], which uses an iterative attention mechanism, has received increasing interest lately with many proposed advancements [37, 38, 47, 62, 68, 71, 75, 81] and extensions to the video domain [14, 22, 41, 69, 87]. Recently, Seitzer et al. [64] proposed to apply Slot Attention to the features of a pretrained DINO model, successfully scaling it to real-world datasets.

In comparison, there has been little work on continuous and distributed approaches for object discovery. A line of work has made use of complex-valued activations with a similar interpretation of magnitude and orientation to our proposed Rotating Features, starting from supervised [51, 89] and weakly supervised methods [58, 59, 60], to unsupervised approaches [48, 61]. However, they are only applicable to very simple, grayscale data. Concurrent work [73] proposes a contrastive learning method for the Complex AutoEncoder, scaling it to simulated RGB datasets of simple, uniformly colored objects. With the proposed Rotating Features, new evaluation procedure and higher-level input features, we create distributed and continuous object-centric representations that are applicable to real-world data.

## 6   Conclusion

**Summary** We propose several key improvements for continuous and distributed object-centric representations, scaling them from toy to real-world data. Our introduction of Rotating Features, a new evaluation procedure, and a novel architecture that leverages pre-trained features has allowed for a more scalable and expressive approach to address the binding problem in machine learning.

**Limitations and Future Work** Rotating Features create the capacity to represent more objects at once. However, this capacity can currently only be fully leveraged in the output space. As we show in Appendix D.2, within the bottleneck of the autoencoder, the separation of objects is not pronounced enough, yet, and we can only extract up to two objects at a time. To overcome this limitation, further research into novel inference methods using Rotating Features will be necessary. In addition, when it comes to object discovery results, Rotating Features produce slightly inferior outcomes than the state-of-the-art DINOSAUR autoregressive Transformer model, while still surpassing standard MLP decoders. Nonetheless, we hope that by exploring a new paradigm for object-centric representation learning, we can spark further innovation and inspire researchers to develop novel methods that better reflect the complexity and richness of human cognition.

## Acknowledgments and Disclosure of Funding

We thank Jascha Sohl-Dickstein and Sjoerd van Steenkiste for insightful discussions. Francesco Locatello worked on this paper outside of Amazon. Sindy Löwe was supported by the Google PhD Fellowship.

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

# Appendix
# Rotating Features for Object Discovery

## A  Broader Impact

This research explores advancements in continuous and distributed object-centric representations, scaling them from simple toy to real-world data. The broader impact includes:

1. Advancing Machine Learning: The proposed techniques expand the methodological landscape for object discovery, which may benefit many areas of machine learning, such as computer vision, robotics, and natural language processing.

2. Biological Insight: By exploring biologically inspired approaches to resolve the binding problem in machine learning, we may advance our understanding of object representations in the brain.

3. Improved Interpretability: The research promotes more structured representations, which may improve interpretability in machine learning models.

4. Scalability and Real-World Applicability: The proposed approaches can be applied to complex, multi-dimensional data, such as RGB images, making continuous object-centric representations applicable in real-world scenarios.

This research inspires further innovation and interdisciplinary collaboration. While we currently do not anticipate any negative societal impacts, we recognize that the wide array of potential applications for object discovery methods could lead to unforeseen consequences. As such, we acknowledge that there may be potential impacts that we are unable to predict at this time.

## B  Reproducibility Statement

In order to ensure the reproducibility of our experiments, we have provided a comprehensive overview of the employed model architectures, hyperparameters, evaluation procedures, datasets and baselines in Appendix C. Additionally, we published the code used to produce the main experimental results at github.com/loeweX/RotatingFeatures.

To achieve stable and reproducible results, we conducted all experiments using 4-5 different seeds. Training and evaluating the roughly 180 Rotating Features models highlighted across all our experiments took approximately 450 hours on a single Nvidia GTX 1080Ti. Training and evaluating four Slot Attention models for the baseline in Section 4.4 took another 200 hours on the same device. This estimate does not take into account the time spent on hyperparameter search and various trials conducted throughout the research process.

## C  Implementation Details

### C.1  Architecture and Hyperparameters

We implement Rotating Features within an autoencoding model. Here, we will describe the precise architecture and hyperparameters used to train this model.

**Architecture**  We provide a detailed description of the autoencoding architecture used for all of our experiments in Table 2. For the activation function (Eq. (4)), we employ Batch Normalization [35] in all convolutional layers, and Layer Normalization [2] in all fully connected layers.

**Parameter Count**  As only the biases depend on the number of rotation dimensions $n$, the number of learnable parameters increases only minimally for larger values of $n$. For instance, in the models utilized for the 4Shapes dataset (Table 2 with $d = 32$), the total number of parameters amounts to 343,626 for $n = 2$ and 352,848 for $n = 8$, reflecting a mere $2.7\%$ increase.

Table 2: Autoencoding architecture with input dimensions $h, w, c$ and feature dimension $d$ (values for $d$ are specified in Tables 3, 4 and 7). All fractions are rounded up. Layers denoted by $*$ are only present in the models that are applied to pretrained DINO features. Otherwise, the feature dimension of the final TransConv layer is $h \times w \times c$.

|  | Layer | Feature Dimension (H $\times$ W $\times$ C) | Kernel | Stride | Padding Input / Output |
|---|---|---|---|---|---|
|  | Input | $h \times w \times c$ |  |  |  |
| Encoder | Conv | $h/2 \times w/2 \times d$ | 3 | 2 | 1 / 0 |
|  | Conv | $h/2 \times w/2 \times d$ | 3 | 1 | 1 / 0 |
|  | Conv | $h/4 \times w/4 \times 2d$ | 3 | 2 | 1 / 0 |
|  | Conv | $h/4 \times w/4 \times 2d$ | 3 | 1 | 1 / 0 |
|  | Conv | $h/8 \times w/8 \times 2d$ | 3 | 2 | 1 / 0 |
|  | Conv* | $h/8 \times w/8 \times 2d$ | 3 | 1 | 1 / 0 |
|  | Reshape | $1 \times 1 \times (h/8 * w/8 * 2d)$ | - | - | - |
|  | Linear | $1 \times 1 \times 2d$ | - | - | - |
| Decoder | Linear | $1 \times 1 \times (h/8 * w/8 * 2d)$ | - | - | - |
|  | Reshape | $h/8 \times w/8 \times 2d$ | - | - | - |
|  | TransConv | $h/4 \times w/4 \times 2d$ | 3 | 2 | 1 / 1 |
|  | Conv | $h/4 \times w/4 \times 2d$ | 3 | 1 | 1 / 0 |
|  | TransConv | $h/2 \times w/2 \times 2d$ | 3 | 2 | 1 / 1 |
|  | Conv | $h/2 \times w/2 \times d$ | 3 | 1 | 1 / 0 |
|  | TransConv | $h \times w \times d$ | 3 | 2 | 1 / 1 |
|  | Conv* | $h \times w \times c$ | 3 | 1 | 1 / 0 |

**Hyperparameters** All hyperparameters utilized for models trained directly on raw input images are detailed in Table 3, while those for models trained on DINO features can be found in Table 4. We initialize the weights $\mathbf{w}_{\text{out}} \in \mathbb{R}^c$ of $f_{\text{out}}$ by setting them to zero, and set biases $\mathbf{b}_{\text{out}} \in \mathbb{R}^c$ to one. The remaining parameters are initialized using PyTorch's default initialization methods.

Table 3: Hyperparameters for models with Rotating Features that are trained directly on the raw input images.

| Dataset | 4Shapes | 4Shapes RGB(-D) | 10Shapes |
|---|---|---|---|
| Training Steps | 100k | 100k | 100k |
| Batch Size | 64 | 64 | 64 |
| LR Warmup Steps | 500 | 500 | 500 |
| Peak LR | 0.001 | 0.001 | 0.001 |
| Gradient Norm Clipping | 0.1 | 0.1 | 0.1 |
| Feature dim $d$ | 32 | 64 | 32 |
| Rotating dim $n$ | 8 | 8 | 10 |
| Image/Crop Size | 32 | 32 | 48 |
| Cropping Strategy | Full | Full | Full |
| Augmentations | - | - | - |
| $k$ | 5 | 5 | 11 |

**ViT Encoder** To generate higher-level input features of real-world images, we use a pretrained vision transformer model [11] to preprocess the input images. Specifically, we employ a pretrained DINO model [5] from the timm library [84], denoted as ViT-B/16 or "vit_base_patch16_224". This model features a token dimensionality of 768, 12 heads, a patch size of 16, and consists of 12 Transformer blocks. We use the final Transformer block's output as input for the Rotating Features model, excluding the last layer norm and the CLS token entry. All input images for the ViT have a resolution of $224 \times 224$ pixels and are normalized based on the statistics of the ImageNet dataset

Table 4: Hyperparameters for models with Rotating Features that are trained on DINO preprocessed features.

| Dataset | Pascal | FoodSeg103 |
|---|---|---|
| Training Steps | 30k | 30k |
| Batch Size | 64 | 64 |
| LR Warmup Steps | 5000 | 5000 |
| Peak LR | 0.001 | 0.001 |
| Gradient Norm Clipping | 0.1 | 0.1 |
| Feature dim $d$ | 128 | 128 |
| Rotating dim $n$ | 10 | 10 |
| Image/Crop Size | 224 | 224 |
| Cropping Strategy | Random | Random |
| Augmentations | Random Horizontal Flip | Random Horizontal Flip and $\pm 90°$ rotation |
| $k$ | 4 | 5 |

[10], with mean per channel (0.485, 0.456, 0.406) and standard deviation per channel (0.229, 0.224, 0.225). The output of the DINO model has a dimension of $14 \times 14 \times 768$ and we train our model to reconstruct this output. Before using it as input, we apply Batch Normalization and ReLU to ensure all inputs to the Rotating Features are positive. Overall, this setup closely follows that of Seitzer et al. [64] to ensure comparability of our results.

## C.2 Evaluation Details

To evaluate the learned object separation of Rotating Features, we cluster their output orientations. For this, we normalize all output features onto the unit-hypersphere and subsequently apply a weighted average that masks features with small magnitudes (Section 3.4). For this masking, we use a threshold of $t = 0.1$ across all experiments. Subsequently, we apply a clustering method to the weighted average. For $k$-means, we utilize the scikit-learn implementation ([55], `sklearn.cluster.KMeans`) with $k$ set to the value as specified in Tables 3, 4 and 7, and all other parameters set to their default values. For agglomerative clustering, we also use the scikit-learn implementation (`sklearn.cluster.AgglomerativeClustering`) with `n_clusters` set to None. The distance thresholds set to 5 for the results in Fig. 9, or a range of values as visualized in Figs. 7 and 22.

When utilizing DINO preprocessed features, our predicted object masks have a size of $14 \times 14$, and we resize them to match the size of the ground-truth labels using the "nearest" interpolation mode. Subsequently, we smooth the mask by applying a modal filter with a disk-shaped footprint of radius $\lfloor 320/14 \rfloor = 22$ using the scikit-image functions `skimage.filters.rank.modal` and `skimage.morphology.disk` [78].

For calculating the final object discovery metrics, we treat unlabeled pixels and pixels from overlapping instances as background (i.e. they are not evaluated).

## C.3 Datasets

**4Shapes** The 4Shapes dataset comprises grayscale images of dimensions $32 \times 32$, each containing four distinct white shapes ($\square, \triangle, \triangledown, \bigcirc$) on a black background. The square has an outer side-length of 13 pixels, the circle has an outer radius of 11 pixels and both isosceles triangles have a base-length of 17 pixels and a height of 9 pixels. All shapes have an outline width of 3 pixels. The dataset consists of 50,000 images in the train set, and 10,000 images for the validation and test sets, respectively. All pixel values fall within the range [0,1].

**4Shapes RGB(-D)** The RGB(-D) variants of the 4Shapes dataset follow the same general setup, but randomly sample the color of each shape. To achieve this, we create sets of potential colors with varying sizes. Each set is generated by uniformly sampling an offset value within the range [0,1], and subsequently producing different colors by evenly dividing the hue space, starting from this offset value. The saturation and value are set to one for all colors, and the resulting HSV color representations are converted to RGB. To create the RGB-D variant, we incorporate a depth channel

to each image and assign a unique depth value within the range [0,1] to every object, maintaining equal distances between them.

**10Shapes** The 10Shapes dataset consists of RGB-D images with $48 \times 48$ pixels. Each image features ten unique shapes ($\square, \triangle, \bigcirc$ in varying orientations and sizes) on a black background. This dataset includes the same square, circle, and triangle as the 4Shapes dataset, with additional variations: the triangle is rotated by $\pm 90°$ degrees to form two more shapes, and the square is rotated by $45°$ to create a diamond shape. Another circle with an outer radius of 19 is also included. All these objects have an outline width of 3 pixels, with their centers remaining unfilled. The dataset also contains two smaller solid shapes: a downward-pointing triangle with a base-length of 11 and a height of 11, and a square with a side-length of 7. Unique colors are assigned to each object by evenly dividing the hue space, starting from an offset value uniformly sampled from the range [0,1]. The saturation and value are set to one for all colors, and the resulting HSV color representations are converted to RGB. Furthermore, each object is given a distinct depth value within the range [0,1], ensuring equal distances between them. The dataset consists of 200,000 images in the train set, and 10,000 images for the validation and test sets, respectively, and all pixel values fall within the range [0,1].

**Pascal VOC** The Pascal VOC dataset [19] is a widely-used dataset for object detection and semantic segmentation. Following the approach of Seitzer et al. [64], we train our model on the "trainaug" variant, an unofficial split consisting of 10,582 images, which includes 1,464 images from the original segmentation train set and 9,118 images from the SBD dataset [30]. During training, we preprocess images by first resizing them such that their shorter side measures 224 pixels, followed by taking a random crop of size $224 \times 224$. Additionally, we apply random horizontal flipping. We evaluate the performance of Rotating Features on the official segmentation validation set, containing 1,449 images. Consistent with previous work [29, 36, 64], we use a resolution of $320 \times 320$ pixels for evaluation. To achieve this, we resize the ground-truth segmentation mask such that the shorter side measures 320 pixels and then take a center crop of dimensions $320 \times 320$. During evaluation, unlabeled pixels are ignored. To make our results comparable to Seitzer et al. [64], we evaluate the performance of our model across 5 seeds on this dataset.

**FoodSeg103** The FoodSeg103 dataset [85] serves as a benchmark for food image segmentation, comprising 4,983 training images and 2,135 test images. We apply the same preprocessing steps as described for the Pascal VOC dataset, with the addition of a random augmentation. Besides randomly horizontally flipping the training images, we also rotate them randomly by $90°$ in either direction.

## C.4 Baselines

**Complex AutoEncoder (CAE)** The CAE [48] is a continuous and distributed object-centric representation learning approach that has been shown to work well on simple toy datasets. To reproduce this model, we make use of its open-source implementation at `https://github.com/loeweX/ComplexAutoEncoder`.

**AutoEncoder** To establish a standard autoencoder baseline, we employ the same architecture as described in Table 2, but replace the Rotating Features and layer structure of $f_{\text{rot}}$ with standard features and layers. Similarly to the Rotating Features model, we apply Batch Normalization / Layer Normalization before the ReLU non-linearity and maintain an input and output range of [0,1].

**Block Masks** To create the block masks baseline, we follow a similar approach as described by Seitzer et al. [64]. First, we divide the image into a specific number of columns: for fewer than 9 blocks, we create 2 columns; for 9 to 15 blocks, we create 3 columns, and for more than 15 blocks, we create 4 columns. Next, we distribute the blocks evenly across the columns. If equal division is not possible, we first allocate blocks to all but the last column and then display the remaining blocks in the final column. See Fig. 10 for examples with 5 and 10 blocks.

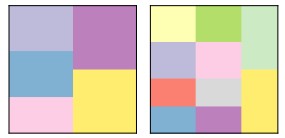

Figure 10: Block masks with 5 and 10 blocks.

**Other Baselines on the Pascal VOC dataset** The results of the Slot Attention, SLATE, DINO $k$-means, DINOSAUR Transformer, DINOSAUR MLP models on the Pascal VOC dataset are adapted from Seitzer et al. [64]. To ensure the comparability of our findings, we adhere as closely as possible to their setup, utilizing the same dataset structure, preprocessing, and DINO pretrained model.

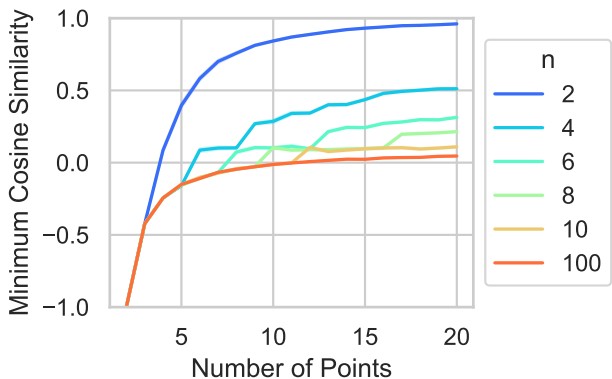

Figure 11: Theoretical investigation into the capacity of Rotating Features to represent many objects simultaneously. We plot the minimal cosine similarity (y-axis) achievable between any pair of points when placing a specific number of points (x-axis) onto an $n$-dimensional hypersphere (mean $\pm$ SEM across 10 seeds). We observe that starting from 10 points and with $n$ large enough, the minimal cosine similarity barely increases. Together with our experiment showing that Rotating Features are able to separate 10 objects, we thus infer that they should be able to scale to larger numbers of objects effectively.

**Slot Attention**   To implement Slot Attention [46], we follow its open-source implementation at `https://github.com/google-research/google-research/tree/master/slot_attention`. We use a hidden dimension of 64 across the model and modulate the decoding structure as outlined in Table 5.

Table 5: Architecture used for the Spatial-Broadcast Decoder in the Slot Attention model.

| Layer | Feature Dimension (H × W × C) | Kernel | Stride | Padding Input / Output | Activation Function |
|---|---|---|---|---|---|
| Spatial Broadcast | 6 × 6 × 64 | - | - | - | - |
| Position Embedding | 6 × 6 × 64 | - | - | - |  |
| TransConv | 12 × 12 × 64 | 5 | 2 | 2 / 1 | ReLU |
| TransConv | 24 × 24 × 64 | 5 | 2 | 2 / 1 | ReLU |
| TransConv | 48 × 48 × 64 | 5 | 2 | 2 / 1 | ReLU |
| TransConv | 48 × 48 × 64 | 5 | 1 | 2 / 0 | ReLU |
| TransConv | 48 × 48 × 5 | 3 | 1 | 1 / 0 | ReLU |

# D   Additional Results

In this section, we provide additional results exploring the theoretical capacity of Rotating Features to represent many objects simultaneously, introducing a strong baseline for object discovery on the Pascal VOC dataset, and showcasing object separation in the latent space of a model trained with Rotating Features. Furthermore, we will provide additional quantitative and qualitative results for the experiments presented in the main paper.

## D.1   High-dimensional Hyperspheres

We explore the theoretical capacity of Rotating Features to represent many objects simultaneously. Rotating Features learn continuous object-centric representations by separating object representations in orientation space. Depending on their orientation, the binding mechanism influences which features are processed together and which ones are masked out. This ultimately affects the number of objects that can be represented separately. Since this mechanism depends on the cosine similarity between features, we can analyze the capacity of Rotating Features by investigating the minimum cosine similarity achievable for a set of points on a hypersphere.

We investigate the behavior of the minimal cosine similarity between points on a hypersphere by implementing an optimization procedure that closely follows Mettes et al. [49]. This is necessary due to the lack of an exact solution for optimally distributing a variable number of points on higher-dimensional hyperspheres [52, 76]. To begin, we randomly sample points on an $n$-dimensional hypersphere and then, for 10,000 steps, iteratively minimize the maximum cosine similarity between any pair of points using Stochastic Gradient Descent (SGD) with a learning rate of 0.1 and momentum of 0.9. In Fig. 11, we display the final results of this optimization process, which indicate the minimal achievable cosine similarity between any pair of points when a certain number of points are positioned on an $n$-dimensional hypersphere.

From this plot, we observe that regardless of $n$, we can always separate two points by a minimum cosine similarity of -1. When $n = 2$, which corresponds to the setting of the Complex AutoEncoder, the minimal distance between points increases rapidly as more points are added. In this setting, when placing 20 points onto the hypersphere, the minimal cosine similarity approaches 1, indicating that points are closely packed together and barely separable from one another. With Rotating Features, we introduce a method to learn features with $n \geq 2$. As evident from the figure, larger values of $n$ enable us to add more points to the hypersphere while maintaining small minimal cosine similarities. Finally, we observe that starting from ten points, if $n$ is sufficiently large, the minimal cosine similarity achievable between points barely increases as more points are added. Having demonstrated that Rotating Features can separately represent ten objects with our experiments on the 10Shapes dataset (Section 4.1), the minimum cosine similarity achievable between 10 points is sufficient for the model to be able to separate them. Thus, we conclude that the number of objects to separate is not a critical hyperparameter for Rotating Features, in contrast to slot-based approaches.

### D.2 Weakly Supervised Semantic Segmentation

In our main experiments, we demonstrate that Rotating Features create the capacity to represent more objects simultaneously (Section 4.1). Here, we explore the object separation within the bottleneck of an autoencoder utilizing Rotating Features. Our aim is to highlight that Rotating Features introduce object-centricity throughout the entire model and to examine whether their increased capacity is utilized in the latent space. To accomplish this, we create a weakly supervised semantic segmentation setup. During the autoencoder training, we introduce a supervised task in which a binary classifier is trained to identify the classes of all objects present in an image. Subsequently, we align the orientations of the class predictions with those of the object reconstructions in order to generate semantic segmentation masks.

**Method**  To train a network with Rotating Features on a classification task, we apply a linear layer $f_{\text{class}}$ following the structure of $f_{\text{rot}}$ to the latent representation $\mathbf{z}_{\text{encoder}} \in \mathbb{R}^{n \times d_{\text{encoder}}}$ created by the encoder. This layer maps the latent dimension of the encoder $d_{\text{encoder}}$ to the number of possible object classes $o$:

$$\mathbf{z}_{\text{class}} = f_{\text{class}}(\mathbf{z}_{\text{encoder}}) \quad \in \mathbb{R}^{n \times o} \tag{10}$$

Then, we extract the magnitude from this output $\|\mathbf{z}_{\text{class}}\|_2 \in \mathbb{R}^o$ and rescale it using a linear layer $f_{\text{out}}$ with sigmoid activation function before applying a binary cross-entropy loss (BCE) comparing the predicted to the ground-truth object labels $\mathbf{o}$:

$$\mathcal{L}_{\text{class}} = \text{BCE}(f_{\text{out}}(\|\mathbf{z}_{\text{class}}\|_2), \mathbf{o}) \quad \in \mathbb{R} \tag{11}$$

We use the resulting loss value $\mathcal{L}_{\text{class}}$ alongside the reconstruction loss $\mathcal{L}$ to train the network while weighting both loss terms equally.

After training, we leverage the object separation within the bottleneck of the autoencoder to assign each individual pixel a label representing its underlying object class. To do so, we first measure the cosine similarity between each pixel in the normalized rotating reconstruction $\mathbf{z}_{\text{eval}} \in \mathbb{R}^{n \times h \times w}$ (Eq. (8)) and each value in $\mathbf{z}_{\text{class}}$ for which $\sigma(\mathbf{z}_{\text{class}}^i) > 0.5$. In other words, we calculate the cosine similarity for the values corresponding to object classes predicted to be present in the current image. Subsequently, we label each pixel based on the class label of the value in $\mathbf{z}_{\text{class}}$ that it exhibits the highest cosine similarity with. If the maximum cosine similarity to all values in $\mathbf{z}_{\text{class}}$ is lower than a specific threshold (zero in our case), or if $\|\mathbf{z}_{\text{eval}}\|_2 < 0.1$, we assign the pixel to the background label. Note that this procedure does not require any clustering method to be employed, but directly utilizes the continuous nature of Rotating Features.

Table 6: Weakly supervised semantic segmentation (WSSS) results (mean ± SEM across 4 seeds). While the latent representations within the bottleneck of an autoencoder trained with Rotating Features can be used to create accurate semantic segmentation masks on the MNIST1Shape dataset in terms of IoU, the same approach fails when adding another object to the input images (MNIST2Shape) despite achieving strong object discovery (ARI-BG) and object classification results (Accuracy). This highlights that Rotating Features can be successfully employed in a weakly supervised semantic segmentation setting, but that the object separation within the bottleneck of the autoencoder is not pronounced enough, yet, to separate more than two objects.

| Dataset | Method | IoU ↑ | ARI-BG ↑ | Accuracy ↑ |
|---|---|---|---|---|
| MNIST1Shape | WSSS | $0.785_{\pm 0.011}$ | $0.926_{\pm 0.007}$ | $0.996_{\pm 0.000}$ |
| | Baseline | $0.393_{\pm 0.006}$ | | |
| MNIST2Shape | WSSS | $0.230_{\pm 0.043}$ | $0.871_{\pm 0.029}$ | $0.991_{\pm 0.000}$ |
| | Baseline | $0.265_{\pm 0.006}$ | | |

**Setup** We construct an autoencoder adhering to the architecture outlined in Table 2, setting $d = 64$. The model is trained for 10,000 steps with a batch size of 64 using the Adam optimizer [40]. The learning rate is linearly warmed up for the initial 500 steps of training to reach its final value of 0.001. This setup is applied to two datasets: MNIST1Shape and MNIST2Shape.

MNIST1Shape contains $32 \times 32$ images, each featuring an MNIST digit and one of three possible shapes ($\square, \triangle, \triangledown$) that share the same format with those in the 4Shapes dataset. Conversely, MNIST2Shape combines an MNIST digit with two of these three shapes per image. Both datasets contain 50,000 train images and 10,000 images for validation and testing.

The network is trained to accurately reconstruct the input image while correctly classifying the MNIST digit, as well as the other shapes present. The setup described above enables the network to produce semantic segmentation masks, despite only being trained on class labels. We assess the performance of this weakly supervised semantic segmentation approach in terms of the intersection-over-union (IoU) score. For comparison purposes, we establish a baseline that generates unlabeled object masks according to the evaluation procedure presented in Section 3.4 using the outputs of the models under evaluation, and randomly assigns each object mask one of the predicted labels. For all results, overlapping areas between objects are masked and ignored during evaluation.

**Results** The results presented in Table 6 and Fig. 12 indicate that on the MNIST1Shape dataset, our approach with Rotating Features achieves strong weakly supervised semantic segmentation performance. It can accurately match objects separated in the latent space to their representations in the output space. Thereby, it almost doubles the performance of the baseline that randomly assigns labels to discovered objects, without requiring any clustering procedure. However, on the MNIST2Shape dataset, the weakly supervised semantic segmentation method falls short (see Table 6 and Fig. 13), even with a relatively large rotation dimension of $n = 6$ and despite achieving strong object discovery and object classification results (in terms of ARI-BG and accuracy scores). This demonstrates that while Rotating Features create the capacity to represent more objects simultaneously, this capacity is not fully utilized within the bottleneck of the autoencoder. Since the object separation is not pronounced enough, the current approach can only reliably separate representations of two objects in the latent space. However, the findings also highlight that Rotating Features create object-centric representations throughout the entire model, as well as a potential new application for Rotating Features in weakly supervised semantic segmentation – an application with intriguing implications for interpretability.

### D.3 Ablation on Chi

We perform an ablation of the binding mechanism. In this analysis, we modify the Rotating Features model by substituting $\mathbf{m}_{\text{bind}}$ with $\|\boldsymbol{\psi}\|_2$ in Eq. (4). This effectively removes the binding mechanism. Then, we apply the adjusted model to the grayscale 4Shapes dataset. While the original model achieves an ARI-BG score of $0.987 \pm 0.003$ on this dataset, the ablated model fails to learn any object separation ($0.059 \pm 0.017$). This result highlights the crucial role the binding mechanism plays in enabling Rotating Features to learn object-centric representations.

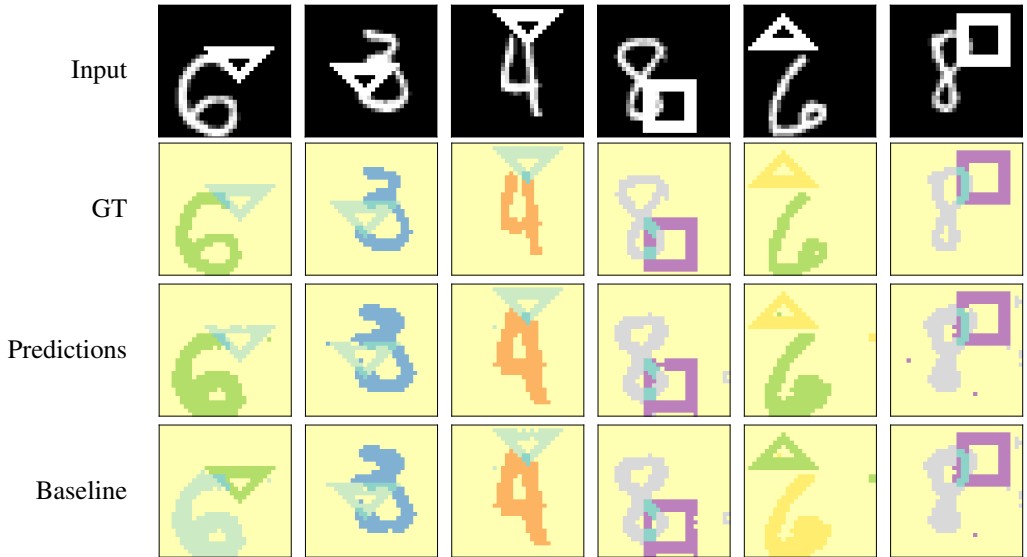

Figure 12: Weakly supervised semantic segmentation results on the MNIST1Shape dataset. Since this is a semantic segmentation task, colors between ground-truth and predicted objects need to match for strong performance. While this is the case for the predictions created by our proposed approach leveraging Rotating Features, the baseline randomly assigns labels to discovered objects, leading to mismatches between predicted and ground-truth object labels.

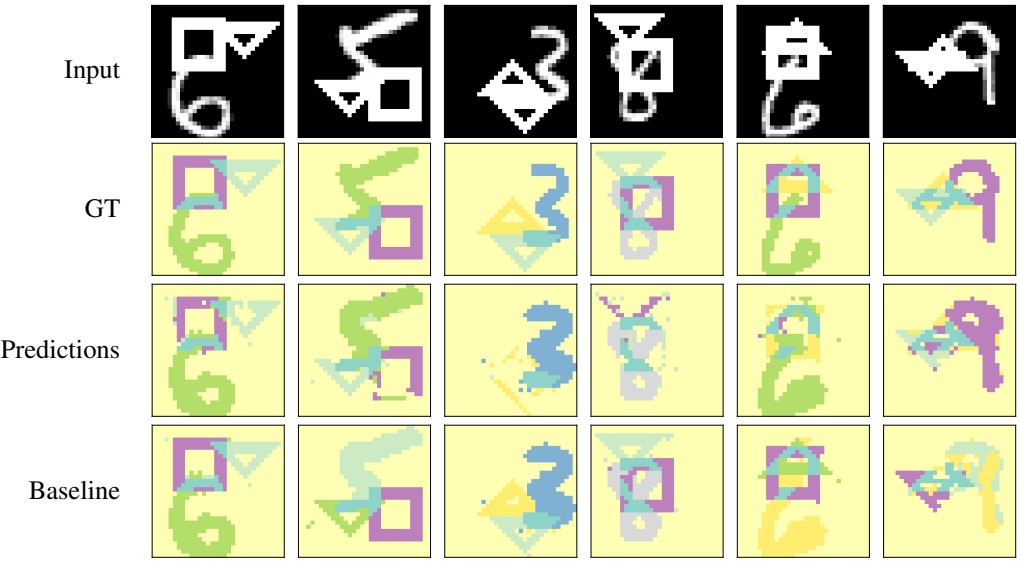

Figure 13: Weakly supervised semantic segmentation results on the MNIST2Shape dataset. The baseline creates accurate object masks, albeit with wrongly predicted labels due to its random matching process. Our proposed approach leveraging Rotating Features, on the other hand, falls short in creating accurate semantic segmentations.

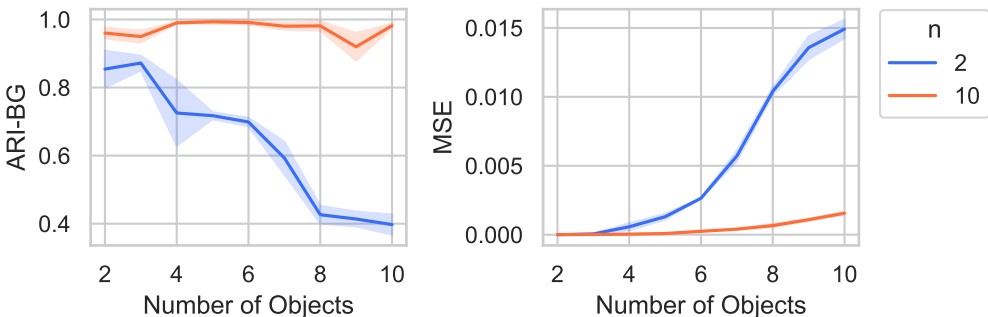

Figure 14: Rotating Features with $n = [2, 10]$ on images containing varying numbers of objects (mean $\pm$ SEM across 4 seeds). As the number of objects increases, the performance of Rotating Features deteriorates strongly when $n = 2$, but remains strong when $n = 10$.

### D.4 ARI-BG on Pascal VOC

We create a baseline that significantly surpasses all previously documented results in the literature in terms of the ARI-BG score on the Pascal VOC dataset. To achieve this, we simply predict the same object label for all pixels in an image, resulting in an ARI-BG score of 0.499. For comparison, the highest reported ARI-BG in the literature for this dataset is 0.296 [81]. We attribute the remarkable performance of this baseline to the characteristics of the Pascal VOC dataset, which often comprises images featuring a large object dominating the scene. Simply predicting the same label across the entire image results in the same label being predicted across this large, central object, and since the ARI-BG score is not evaluated on background pixels, this leads to relatively high scores. However, for the MBO scores, this baseline yields results far below the state-of-the-art, with 0.205 in $\text{MBO}_i$ and 0.247 in $\text{MBO}_c$. Consequently, we focus our evaluation on these scores for this dataset.

### D.5 Rotation Dimension $n$ vs. Number of Objects

We conduct an experiment to assess how the choice of $n$ impacts the object discovery performance of a Rotating Features model for different numbers of objects per scene. We accomplish this by creating variations of the 10Shapes dataset that contain between two and ten objects. For each variant, the number of objects per image equals the total number of distinct objects throughout the respective dataset. As illustrated in Fig. 14, the object discovery performance significantly drops as the number of objects increases when $n = 2$. However, the performance remains consistently high when $n = 10$. These findings indicate that the choice of $n$ becomes increasingly important as the number of objects rises.

### D.6 Multi-dSprites and CLEVR

We evaluate the performance of the proposed Rotating Features model on the commonly used Multi-dSprites and CLEVR datasets [39], comparing it to its predecessor, the CAE model [48]. The hyperparameters are listed in Table 7 and the outcomes are presented in Table 8. Note that since the original CAE was only applicable to grayscale images, we combine it with our proposed evaluation procedure to make it applicable to multi-channel input images, which we denote as CAE*.

The results indicate that the Rotating Features model significantly surpasses the performance of the CAE*, demonstrating good object separation on both datasets. However, the results still lack behind the state-of-the-art. The qualitative examples in Figs. 15 and 16 show that Rotating Features encounter the same issue here, as we have seen with the colored 4Shapes dataset (Section 4.2): objects of the same color tend to be grouped together. As demonstrated with the RGB-D version of the colored 4Shapes dataset and with our results using pretrained DINO features on real-world images, this issue can be addressed by using higher-level input features.

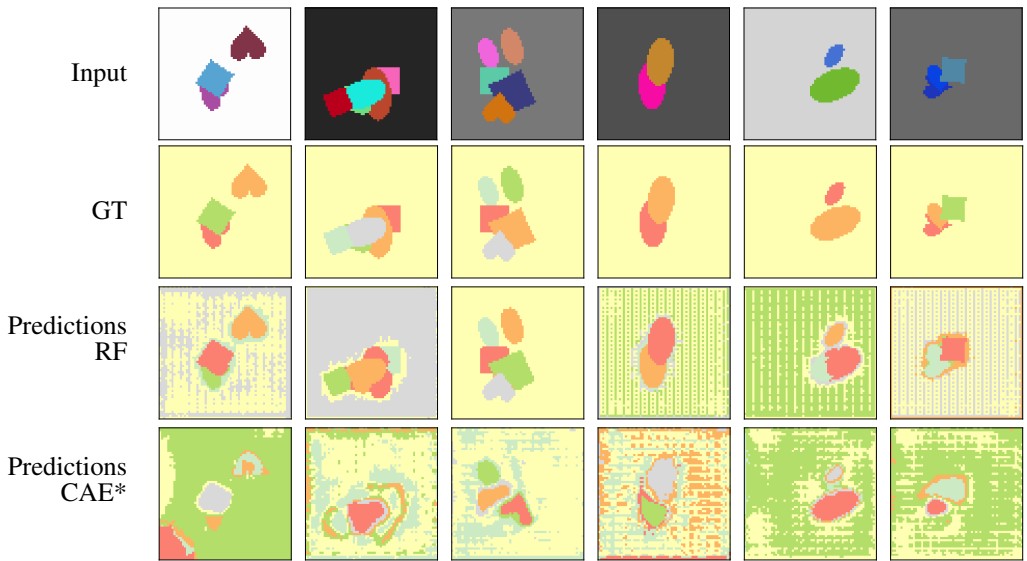

Figure 15: Qualitative performance comparison on the 4Shapes dataset. Rotating Features (RF) create a substantially better object separation than the CAE*, but fail to separate objects of the same color (see right-most column).

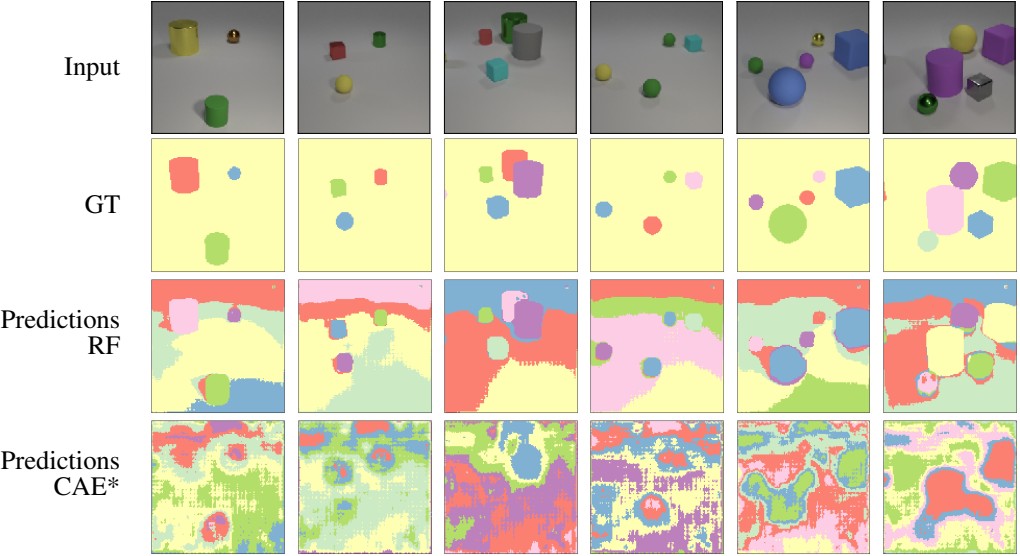

Figure 16: Qualitative performance comparison on the 4Shapes dataset. In contrast to the CAE*, the Rotating Features (RF) model learns to separate objects well. However, it fails to separate objects of the same color (see last three columns).

Table 7: Hyperparameters for the Rotating Features models that are trained on the Multi-dSprites and CLEVR datasets.

| Dataset | Multi-dSprites | CLEVR |
|---|---|---|
| Training Steps | 100k | 30k |
| Batch Size | 16 | 64 |
| LR Warmup Steps | 5000 | 5000 |
| Peak LR | 0.001 | 0.001 |
| Gradient Norm Clipping | 0.1 | 0.1 |
| Feature dim $d$ | 128 | 128 |
| Rotating dim $n$ | 10 | 10 |
| Image/Crop Size | 64 | 128 |
| Cropping Strategy | Random | Random |
| Augmentations | - | - |
| $k$ | 6 | 5 |

Table 8: Performance of Rotating Features and CAE* on the Multi-dSprites and CLEVR datasets. While the CAE* largely fails to separate objects in these datasets, Rotating Features achieve strong results in terms of ARI-BG and $MBO_i$.

| Dataset | Model | MSE $\downarrow$ | ARI-BG $\uparrow$ | $MBO_i \uparrow$ |
|---|---|---|---|---|
| Multi-dSprites | CAE* | 3.540e-03 $\pm$ 1.990e-04 | 0.371 $\pm$ 0.056 | 0.426 $\pm$ 0.042 |
| | Rotating Features | 7.082e-04 $\pm$ 1.173e-04 | 0.888 $\pm$ 0.015 | 0.863 $\pm$ 0.011 |
| CLEVR | CAE* | 1.191e-03 $\pm$ 2.998e-05 | 0.289 $\pm$ 0.042 | 0.234 $\pm$ 0.074 |
| | Rotating Features | 6.643e-04 $\pm$ 5.564e-05 | 0.664 $\pm$ 0.013 | 0.608 $\pm$ 0.017 |

### D.7 Additional Results

We provide supplementary qualitative and quantitative results for the experiments discussed in the main part of this paper. For the 4Shapes, 4Shapes RGB(-D), and 10Shapes datasets, additional qualitative results can be found in Figs. 17 to 19, with corresponding quantitative results detailed in Tables 9 to 11. Additional qualitative results for the Pascal VOC dataset are presented in Fig. 20, and extra results for the FoodSeg103 dataset are provided in Fig. 21. Lastly, we compare the performance of Rotating Features on the FoodSeg103 dataset using $k$-means and agglomerative clustering with various hyperparameters in Fig. 22.

Table 9: Detailed quantitative performance comparison on the 4Shapes (mean $\pm$ SEM across 4 seeds). As the number of rotation dimensions $n$ in the Rotating Features increases, both reconstruction (MSE) and object discovery (ARI-BG and MBO$_i$) performance improve. This allows Rotating Features to significantly outperform the standard autoencoder (AE) and Complex AutoEncoder (CAE).

| Model | $n$ | MSE $\downarrow$ | ARI-BG $\uparrow$ | MBO$_i$ $\uparrow$ |
|---|---|---|---|---|
| AE | - | 5.492e-03 $\pm$ 9.393e-04 | - | - |
| CAE | - | 3.435e-03 $\pm$ 2.899e-04 | 0.694 $\pm$ 0.041 | 0.628 $\pm$ 0.039 |
| | 2 | 2.198e-03 $\pm$ 1.110e-04 | 0.666 $\pm$ 0.046 | 0.589 $\pm$ 0.026 |
| | 3 | 1.112e-03 $\pm$ 2.698e-04 | 0.937 $\pm$ 0.018 | 0.861 $\pm$ 0.020 |
| | 4 | 5.893e-04 $\pm$ 4.350e-05 | 0.944 $\pm$ 0.016 | 0.908 $\pm$ 0.012 |
| Rotating | 5 | 5.439e-04 $\pm$ 6.984e-05 | 0.975 $\pm$ 0.003 | 0.934 $\pm$ 0.006 |
| Features | 6 | 2.526e-04 $\pm$ 1.416e-05 | 0.991 $\pm$ 0.002 | 0.970 $\pm$ 0.003 |
| | 7 | 1.642e-04 $\pm$ 1.810e-05 | 0.992 $\pm$ 0.002 | 0.974 $\pm$ 0.003 |
| | 8 | 1.360e-04 $\pm$ 7.644e-06 | 0.987 $\pm$ 0.003 | 0.968 $\pm$ 0.008 |

Table 10: Detailed results on the 4Shapes RGB(-D) datasets (mean $\pm$ SEM across 4 seeds). As we increase the number of potential colors that objects may take on, the performance of Rotating Features worsens on the RGB dataset. However, this effect can be mitigated when providing depth information for each object (RGB-D).

| Dataset | Number of colors | MSE $\downarrow$ | ARI-BG $\uparrow$ | MBO$_i$ $\uparrow$ |
|---|---|---|---|---|
| RGB | 1 | 1.017e-05 $\pm$ 2.541e-06 | 0.982 $\pm$ 0.009 | 0.874 $\pm$ 0.008 |
| RGB | 2 | 1.597e-04 $\pm$ 2.185e-05 | 0.726 $\pm$ 0.035 | 0.595 $\pm$ 0.015 |
| RGB | 3 | 1.044e-04 $\pm$ 6.547e-06 | 0.850 $\pm$ 0.008 | 0.767 $\pm$ 0.016 |
| RGB | 4 | 1.166e-04 $\pm$ 5.985e-06 | 0.682 $\pm$ 0.041 | 0.626 $\pm$ 0.050 |
| RGB | 5 | 1.653e-04 $\pm$ 3.065e-05 | 0.494 $\pm$ 0.056 | 0.474 $\pm$ 0.031 |
| RGB-D | 1 | 4.920e-06 $\pm$ 5.421e-07 | 1.000 $\pm$ 0.000 | 0.875 $\pm$ 0.015 |
| RGB-D | 2 | 4.693e-05 $\pm$ 2.166e-05 | 0.996 $\pm$ 0.004 | 0.798 $\pm$ 0.007 |
| RGB-D | 3 | 2.436e-05 $\pm$ 3.930e-06 | 0.936 $\pm$ 0.029 | 0.785 $\pm$ 0.017 |
| RGB-D | 4 | 4.562e-05 $\pm$ 4.567e-06 | 0.909 $\pm$ 0.039 | 0.759 $\pm$ 0.013 |
| RGB-D | 5 | 8.850e-05 $\pm$ 3.780e-05 | 0.922 $\pm$ 0.040 | 0.727 $\pm$ 0.014 |

Table 11: Detailed quantitative results on the 10Shapes dataset (mean $\pm$ SEM across 4 seeds). When $n$ is sufficiently large, Rotating Features learn to separate all ten objects, highlighting their capacity to separate many objects simultaneously.

| $n$ | MSE | ARI-BG | MBO$_i$ |
|---|---|---|---|
| 2 | 1.672e-02 $\pm$ 7.893e-04 | 0.341 $\pm$ 0.030 | 0.169 $\pm$ 0.013 |
| 10 | 1.918e-03 $\pm$ 1.315e-04 | 0.959 $\pm$ 0.022 | 0.589 $\pm$ 0.039 |

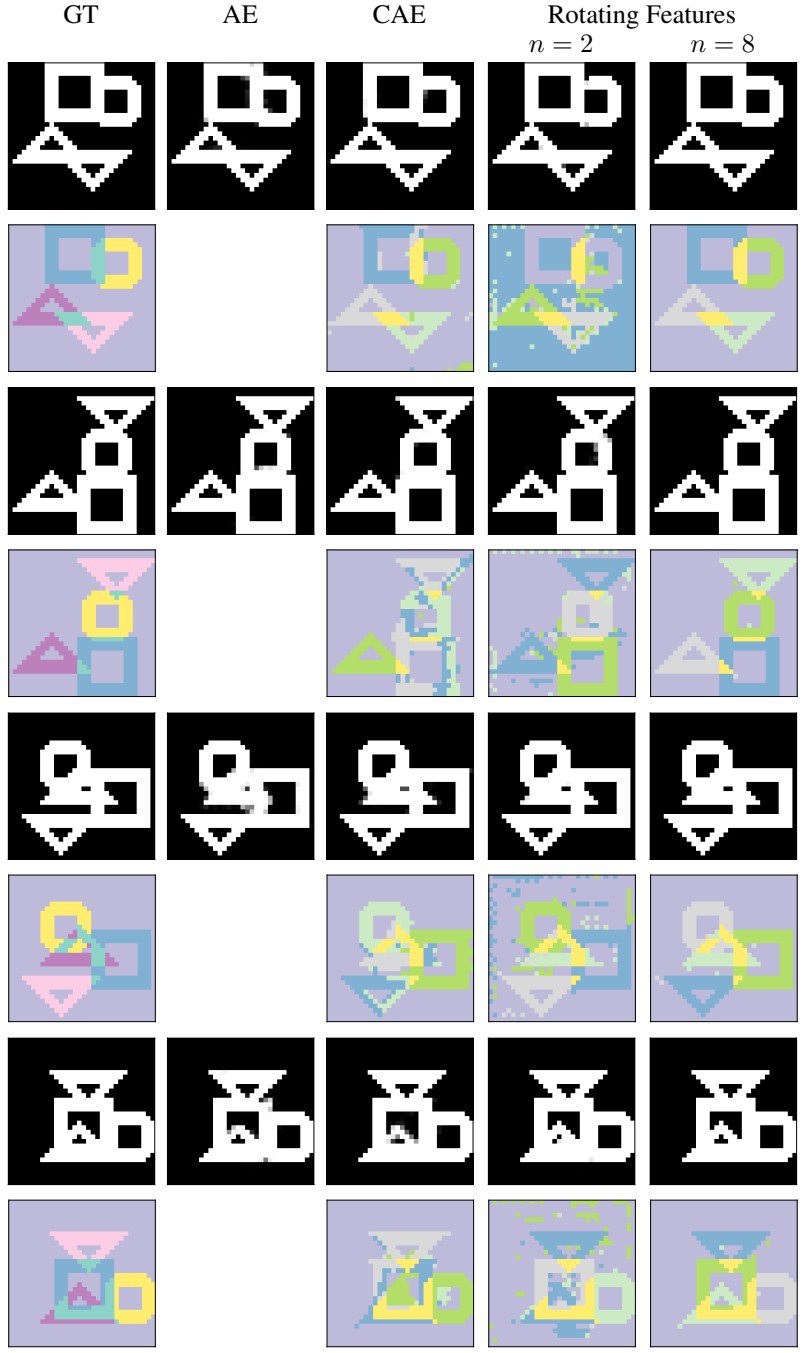

Figure 17: Qualitative performance comparison on the 4Shapes dataset. Rotating Features with a sufficiently large rotation dimension $n$ create sharper reconstructions and significantly better object separations, despite similar numbers of learnable parameters.

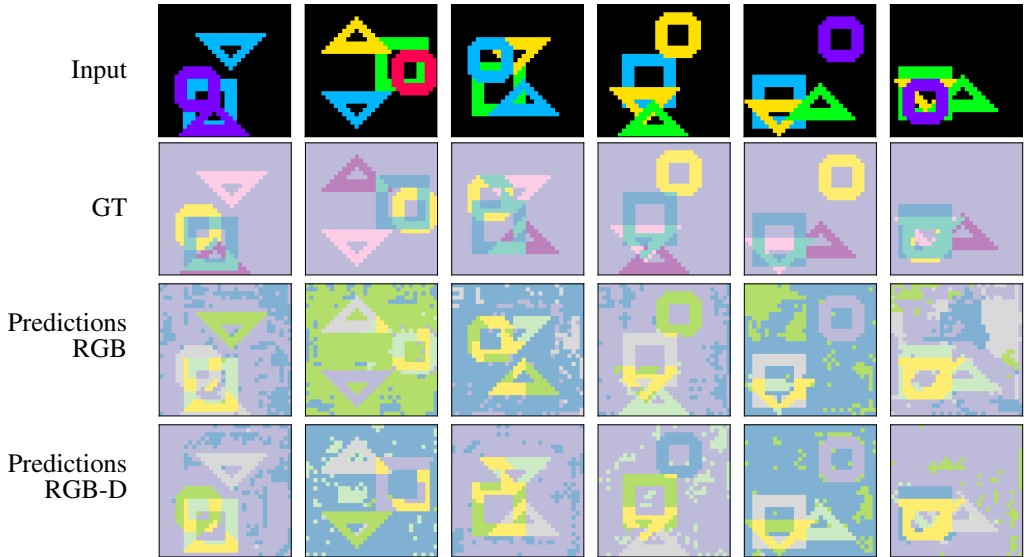

Figure 18: Rotating Features applied to the 4Shapes RGB(-D) datasets. While the model fails to create accurate object separations when applied to the RGB version of this dataset, it can accurately distinguish objects when given additional depth information for each object (RGB-D). Nonetheless, it tends to group one of the four objects with the background.

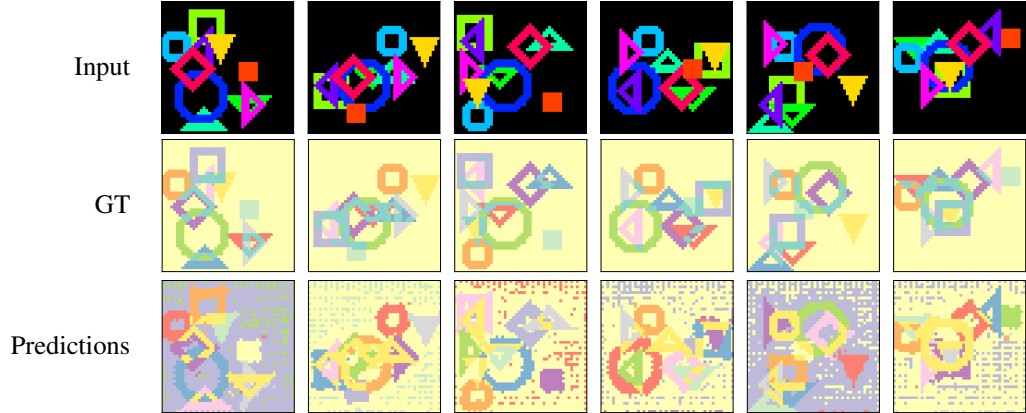

Figure 19: Rotating Features applied to the 10Shapes dataset. With a sufficiently large rotation dimension $n$, Rotating Features are capable to separate all ten objects in this dataset, highlighting their increased capacity to represent many objects simultaneously.

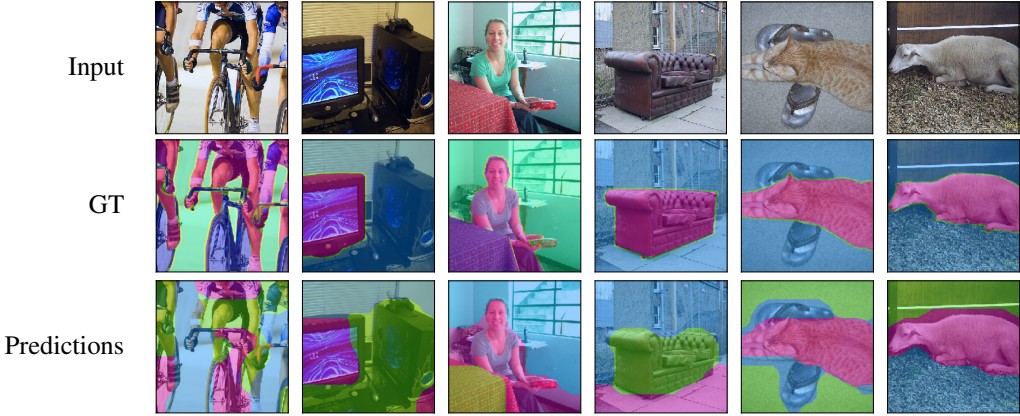

Figure 20: Rotating Features applied to the Pascal VOC dataset. By leveraging higher-level input features created by a pretrained vision transformer, Rotating Features can learn to separate objects in real-world images.

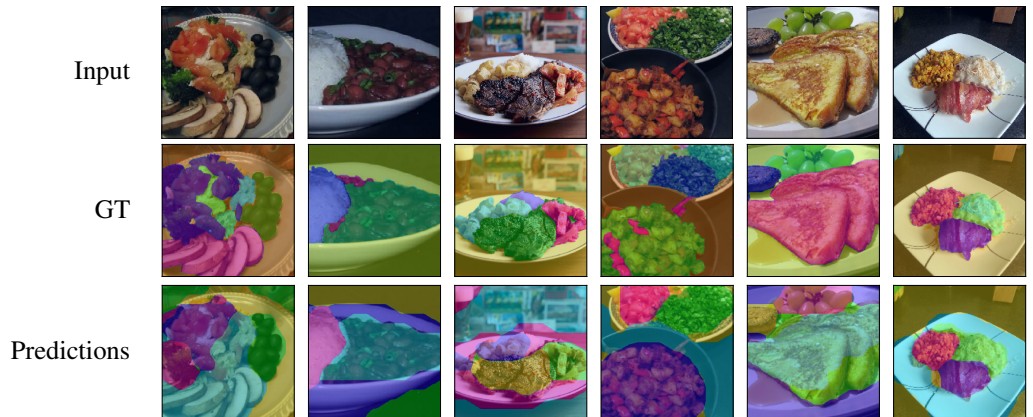

Figure 21: Rotating Features applied to the FoodSeg103 dataset. By utilizing input features generated by a pretrained vision transformer, Rotating Features can learn to distinguish objects in real-world images.

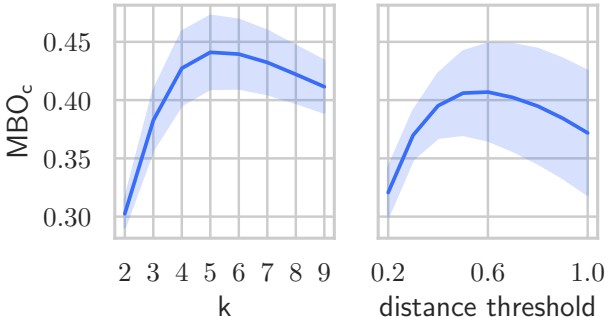

Figure 22: Clustering methods applied to Rotating Features after training on the FoodSeg103 dataset (mean $\pm$ SEM across 4 seeds). Since Rotating Features automatically learn to separate the appropriate amount of objects, we can test various clustering methods and hyperparameters efficiently after training. While agglomerative clustering (right) achieves slightly worse results than $k$-means (left), it eliminates the requirement to set the number of objects to be separated.

