# OpenReview forum: "Rotating Features for Object Discovery"
_NeurIPS.cc/2023/Conference — NeurIPS 2023 oral_

### Official Review · Reviewer_VTkG · 2023-06-30

**Soundness:** 4 excellent
**Presentation:** 3 good
**Contribution:** 3 good
**Rating:** 7
**Confidence:** 4

**Summary:**

CAEs promise to resolve some of the concerns of slot-based representation. They bring the promise of flexible object granularity, the promise of extracting part-whole hierarchies as per need, and faster training speeds. However, the original CAE was tested with grayscale images and on a rather small number of objects (2-3). This paper asks the question, *how can we scale CAEs* beyond this.

**Approach.** Compared to classical neural networks, in the proposed model, all activations in the network are “tagged” with the binding information. For this, every scalar value of conventional vector representation is replaced with a $N$ length vector: the magnitude of this vector plays the role of classical activation while its orientation plays the role of capturing the binding information. Compared to original CAEs, the proposed model provides a generalization from having just 2 components (real and imaginary) to $N$ components, where $N$ can be arbitrarily large.

In **experiments**,
1. The paper tests whether this change can help improve the capacity of CAEs in terms of the number of objects or not.
2. The paper is also one of the first works to scale CAEs to real scenes.
3. The paper also proposes a new procedure for extracting object segments from distributed representations.

**Strengths:**

1. (One of the) First successful attempts to make a CAE-like model scale to more objects and beyond simplistic grayscale images.
2. The way weights and biases were applied seems to have been simplified (a welcome change) compared to the original CAEs. An ablation experiment highlighting this specific change for the case of $N=2$ could be useful though.
3. Seems applicable to any data modality— demonstrated to some extent via the applicability of the model on RGB-D data in addition to RGB data.

**Weaknesses:**

1. One of the novelty seems to be the use of DINO pre-trained features for the first time in the context of CAEs. Therefore, a comparison with and without DINO pre-training would be useful to show in the main paper.
2. Can original CAE be a baseline in the real-data experiment?
3. The paper makes the jump from 4 object scenes directly to 10 object scenes. It may be useful to test perhaps a more gradual increase e.g., by also testing 7 object scenes and 13 object scenes, whether it places gradually more demand on the choice of $N$.
4. It would be interesting to see the performance on a standard dataset like Tetris or CLEVR that was not deliberately designed for this paper. That being said, it need not outperform the previous slot-based methods here considering various other potential benefits of CAEs.
5. At test time, can it handle a larger number of objects than shown during training?

**Questions:**

1. Is the layer choice important for segmentation? Do these patterns emerge also for the intermediate layers?
2. I am curious what is the incentive for the model to assign different “phases” to different objects? In slot-based models, this role was played by the inherent capacity bottleneck of slots. I am curious about some insights about this.
3. I could not entirely follow the motivation of Appendix D.3 “Object separation within the bottleneck of autoencoder utilizing Rotating Features…”

**Limitations:**

Yes, the paper discusses the limitations, but one should also consider the potential benefits of the CAE framework.

---

> ### Author Rebuttal · Authors · 2023-08-09
>
> Thank you for the thoughtful review. We are taking this opportunity to address the concerns and inquiries raised.
>
> ### Strengths
> **2) The way weights and biases were applied seems to have been simplified (a welcome change). An ablation experiment highlighting this specific change for the case of n=2 could be useful.**
>
> Figure 4 in the paper shows that when the CAE model is compared to Rotating Features with $n=2$, they achieve equivalent object discovery performances. This suggests that the modified rotation mechanism does not impact the performance significantly.
>
> ### Weaknesses
>
> **1/2) A comparison with and without DINO pre-training would be useful to show in the main paper. / Can original CAE be a baseline in the real-data experiment?**
>
> In line with the reviewer's suggestion, we ran two additional baselines on the Pascal VOC dataset. We firstly applied the Rotating Features model directly to the raw input images, referred to as RF-DINO. Secondly, we applied a CAE model to the DINO preprocessed features, while incorporating our novel evaluation procedure (CAE* +DINO).
>
> |Model|MBO$_i$|MBO$_c$|
> |--|--|--|
> |RF-DINO|0.282$\pm$0.006|0.320$\pm$0.006|
> |CAE* +DINO|0.329$\pm$0.009|0.374$\pm$0.010|
> |RF+DINO |0.407$\pm$0.001|0.460$\pm$0.001|
>
> The results reveal that our proposed approach of applying Rotating Features to DINO features (RF+DINO) significantly outperforms both baselines. This highlights the importance of our contributions: generalizing the complex-valued features to higher dimensions and using DINO features as the input to our model are both essential to achieving competitive object discovery performance on real-world images. We will include these baselines in the revised paper.
>
> **3) The paper makes the jump from 4 object scenes directly to 10 object scenes. It may be useful to test a more gradual increase.**
>
> In response to the reviewer's suggestion, we conduct a new experiment to assess how the choice of $n$ impacts the object discovery performance of a Rotating Features model depending on the number of objects in a scene. We accomplish this by creating variations of the 10Shapes dataset that contain between two and ten objects. For each variant, the number of objects per image equals the total number of distinct objects throughout the respective dataset.
>
> As illustrated in Figure 2 in the PDF uploaded alongside the general response, the object discovery performance significantly drops as the number of objects increase when $n=2$. However, the performance remains consistently high when $n=10$. These findings indicate that the choice of $n$ becomes increasingly critical as the number of objects rises. We will include this experiment in the revised paper.
>
> **4) It would be interesting to see the performance on a standard dataset.**
>
> In response to the reviewer's suggestion, we evaluate the performance of the proposed Rotating Features model on the Multi-dSprites and CLEVR datasets, comparing it to its predecessor — the CAE model [2]. The outcomes are presented in the table below. Note that since the original CAE was only applicable to grayscale images, we combine it with our proposed evaluation procedure to make it applicable to multi-channel input images, which we denote as CAE*.
>
> |Dataset|Model|ARI-BG|
> |--|--|--|
> |Multi-dSprites|CAE*|0.371$\pm$0.056|
> ||Rotating Features|0.888$\pm$0.015|
> |CLEVR|CAE*|0.289$\pm$0.042|
> ||Rotating Features|0.664$\pm$0.013|
>
> The results indicate that the Rotating Features model significantly surpasses the performance of the CAE*, demonstrating good object separation on these two datasets. However, the results still lack behind the state-of-the-art. The qualitative examples in Figure 3 of the PDF uploaded alongside the general response show that Rotating Features encounter the same issue here, as we have seen with the colored 4Shapes dataset: objects of the same color tend to be grouped together. As demonstrated with the RGB-D version of the colored 4Shapes dataset and our results using pretrained DINO features on real-world images, this issue can be addressed by using higher-level input features.
>
> **5) At test time, can it handle a larger number of objects than shown during training?**
>
> See the general response above.
>
> ### Questions
>
> **1) Is the layer choice important for segmentation? Do these patterns emerge also for the intermediate layers?**
>
> For object segmentation using Rotating Features, the current architectural layout suggests the output layer's representation as the most logical choice, as it has the highest spatial resolution.
>
> Nevertheless, it is interesting to note that object-centric features also emerge in the intermediate layers of the architecture. In Appendix D.3, we examine the object separation within the autoencoder's bottleneck. This experiment reveals that it is possible to distinguish the representations of two separate objects within this layer's representation.
>
> **2) What is the incentive for the model to assign different phases to different objects?**
>
> The binding mechanism enables the model to process information relatively independently of one another. If features of one object have orientations pointing in a different direction than features of another object, their features can be processed with little interference. This mechanism encourages the model to assign different orientations to features that it aims to process separately, which naturally leads to object-centric representations.
>
> **3) I could not entirely follow the motivation of Appendix D.3.**
>
> This experiment is related to question 1 above, and investigates whether object separation also emerges in intermediate layers. Since it is not practical to evaluate the object segmentation performance here, we develop an alternative way to investigate object-centricity. Our approach in this case is a weakly-supervised semantic segmentation setup, which tests the alignment between the intermediate object representations and their counterparts at the output layer.

---

> > ### Comment · Reviewer_VTkG · 2023-08-17
> > **Thank You**
> >
> > Thank you for the rebuttal. The results on CLEVR seem much better relative to CAE and would be nice to include in the paper IMO. I also appreciate testing a gradually increasing number of objects.
> >
> > I change my rating to 7.

---

### Official Review · Reviewer_cH5o · 2023-07-05

**Soundness:** 4 excellent
**Presentation:** 4 excellent
**Contribution:** 4 excellent
**Rating:** 8
**Confidence:** 4

**Summary:**

This work proposes a novel approach to unsupervised object discovery that does not depend on slots. Instead, the model uses an extra set of dimensions to code object assignment based on rotation, potentially allowing for a more flexible distributed form of object discovery than in standard slot-based approaches. The model is shown to perform well on toy tasks, with promising results on real-world images.

**Strengths:**

- This work presents an interesting new direction, substantially reconsidering the problem of object discovery relative to the now ubiquitous slot-based methods.
- The method performs well on toy tasks, scaling to a relatively large number of objects, and also shows promising results on real-world images.
- The method is significantly more efficient than popular slot-based methods.
- The paper stimulates many interesting directions for future work.

**Weaknesses:**

I have a number of suggestions and questions that may help to further improve the paper:

- A major potential advantage of the proposed approach is that it is more flexible than slot-based methods, specifically regarding the number of objects that can be segmented. The results in supplementary figure 10 suggest that, with a sufficient number of dimensions (e.g. ~10) a relatively large number of objects can be represented, and that even more objects can be represented without needing to add many more dimensions. Can this advantage over slot attention be empirically demonstrated? For instance, are there any instances in pascal VOC or FoodSeg that involve more objects than the number of slots in DINOSAUR? If so, it would be interesting to see whether rotating features outperforms DINOSAUR on those problems. Alternatively, controlled experiments with CLEVR, or even the Nshapes dataset, could be performed to demonstrate the superior representational capacity of rotating features over slot attention (especially when there are more objects than slots).
- A desirable feature of slot attention is that it is permutation invariant, which allows for a dynamic binding of features to objects (i.e. by randomly initializing the slots and iteratively refining them through competition). Here, by contrast, particular features have learned biases toward particular orientations, does this interfere with the ability of the model to perform variable-binding in a dynamic manner? How would the model perform when tested on a greater number of objects than it was trained on (given that it has a sufficient number of dimensions to represent those objects)? In other words, is the method more efficient than slot attention because it is also less flexible?
- Have the authors considered sharing the orientation across features at each location in a convolutional feature map? Intuitively, it seems that all features at a particular location should only be assigned to a single object, rather than having a unique assignment of each feature.
- Have the authors investigated how the assignment of objects evolves across layers? I am wondering whether the competition that occurs over time in methods like slot attention is somehow distributed across layers in this model.
- It would be informative to include an ablation of the binding mechanism. I also found the description of this mechanism somewhat counterintuitive. It is described as 'weakening the connections between features with dissimilar orientations', but it almost seems as if it does the exact opposite of this. The magnitude of the features in $\chi$ is completely unrelated to the orientations of the inputs. Therefore, it seems that mixing $\chi$ with $\psi$ is only weakening the influence of orientation, allowing incoming features with dissimilar orientations (but high synaptic weight values) to exercise a greater influence on the magnitude of the feature representation. Can the authors add some additional explanation of this mechanism?

**Questions:**

I have listed some questions in the previous section.

**Limitations:**

There are no discernible negative societal impacts related to this work.

---

> ### Author Rebuttal · Authors · 2023-08-09
>
> Thank you for your insightful review and constructive feedback. We welcome the opportunity to address the questions and concerns that have been brought up.
>
> **1 / 2) How would the model perform when tested on a greater number of objects than it was trained on? / Controlled experiments could be performed to demonstrate the superior representational capacity of rotating features over slot attention (especially when there are more objects than slots).**
>
> See the general response above.
>
> **3) Have the authors considered sharing the orientation across features at each location in a convolutional feature map?**
>
> We have experimented with this, but in preliminary results it generally performed worse. Intuitively, we believe it is helpful to have the possibility for each individual feature to have its own orientation (and therefore object binding), as this allows overlapping objects to be represented simultaneously within the same location. This becomes increasingly important as we reduce the feature map size in the architecture, and essential as we move through the fully-connected layers.
>
> **4) Have the authors investigated how the assignment of objects evolves across layers? I am wondering whether the competition that occurs over time in methods like slot attention is somehow distributed across layers in this model.**
>
> This hypothesis could indeed be correct. In Appendix D.3, we have included an experiment that examines the object separation within the architecture's bottleneck (i.e., after the encoder). While we find a meaningful separation here, it does not seem to be prominent enough, yet, to be able to separate more than two objects at a time. Exploring whether this is indeed a result of some form of competition distributed across layers, and how to modify the network to achieve stronger object separation throughout the entire architecture, would be an intriguing direction for future research.
>
> **5) It would be informative to include an ablation of the binding mechanism. I also found the description of this mechanism somewhat counterintuitive.**
>
> Following the reviewer's suggestion, we performed an ablation of the binding mechanism. In this analysis, we modify the Rotating Features model by substituting $\mathbf{m}_{\text{bind}}$ with $\left\lVert{\psi}\right\rVert_2$ in Equation 4. This effectively removes the binding mechanism ($\chi$). Then, we apply the adjusted model to the grayscale 4Shapes dataset. While the original model achieves an ARI-BG score of $0.987 \pm 0.003$ on this dataset, the ablated model fails to learn any object separation ($0.059 \pm 0.017$). This result highlights the critical role the binding mechanism plays in enabling the Rotating Features to learn object-centric representations. We will include this experiment in the revised paper.
>
> The model without binding mechanism fails to learn object-centric representations, as it cannot leverage the additional rotation dimensions. Without the binding mechanism, these dimensions inherently do not have a strong effect on the computations. The binding mechanism ensures that features with similar orientations are processed together, while features with dissimilar orientations are essentially masked out. This allows the network to create separate streams of information that it can process separately, which naturally leads to the emergence of object-centric representations.
>
> To expand on this intuition, imagine the most extreme scenario where a feature is of the opposite orientation to a group of aligned features, as shown on the left-hand side of Figure 3 (cosine similarity = -1). Without the binding mechanism, the misaligned feature would effectively be subtracted from the aligned features, resulting in a smaller output magnitude (as shown for $\left\lVert{\psi}\right\rVert_2$). The binding mechanism reduces this effect and results in a larger output magnitude (as shown for $\mathbf{m}_{\text{bind}}$). Effectively, the binding mechanism masks out the misaligned feature, as the output magnitude would be the same if the misaligned feature was replaced by a zero vector. We will amend our description in the paper to include this intuition.

---

> > ### Comment · Reviewer_cH5o · 2023-08-16
> > **Reply to rebuttal**
> >
> > Thanks very much to the authors for these clarifications and additional experiments. The explanations and additional results for points 3-5 helped me to develop a better intuition for the method, and I think they will strengthen the paper. My only remaining question is about whether a comparison with slot attention can be performed for the new experiment (detailed in my reply to the general response).

---

> > > ### Comment · Reviewer_cH5o · 2023-08-18
> > >
> > > Thank you to the authors for so thoroughly engaging in the discussion process. All of my concerns have been addressed. I am raising my score to an 8.

---

### Official Review · Reviewer_r4Rq · 2023-07-06

**Soundness:** 3 good
**Presentation:** 3 good
**Contribution:** 2 fair
**Rating:** 6
**Confidence:** 5

**Summary:**

This work addresses the problem of unsupervised object discovery. It seeks to remedy some limitations (primarily object storage capacity) of the recently introduced synchrony-based approach, CAE [1], and scales it to more visually complex scenes compared to CAE [1] which was only applied to simple grayscale (Shapes and MNIST) datasets. The key idea behind the proposed method is the extension of the number of feature dimensions used to manipulate “phase” (rotation) values associated with image features from ‘2’ in CAE to ‘n’ dimensions in RF. To control these ‘n’ dimensional rotation values use ‘n’ separate bias terms in every layer. Further, they apply their method on pre-trained DINO features to scale the grouping results to real-world datasets.


**Strengths:**

1) The problem studied in this paper is well-motivated and important.

2) The method presented is novel and as the model class it uses to perform binding (synchrony-based) has received significantly less attention in the literature compared to slot-based approaches.

3) The paper is very well written.

4) The experiments are well performed and the presented method compares favorably to the considered baselines.


**Weaknesses:**

1) The authors note that one of their three contributions is a new evaluation method. The phases are weighted but with binary values (0 or 1) which reduces to a difference in the averaging constant (N vs N-k) in the denominator of the weighted averaging operation. The threshold used to compute the weights is just a fixed value (i.e. 0.1 same as in CAE [1]) as opposed to being some learned value based on the magnitudes. So, in practice, how different/novel is this evaluation procedure compared to the one used by CAE to the extent that it can be deemed as a core technical contribution of this work?

2) Depth masks provide a strong supervision signal for discovery of object grouping information. The problem of 2 (or more) objects having the same color being grouped together has been resolved through the use of such depth masks i.e. strong supervision (Figure 6 caption).

3) Lines (244-245): “On FoodSeg103, we limit our evaluation of our model to the MBO_c score, …. ”. MBO_c score only measures semantic grouping not instance-level grouping. The pretrained DINO features already show a high level of specialization to semantic classes (attention maps when conditioning on CLS token). Therefore, this experiment does not really meaningfully test the instance-level grouping ability of the proposed method.


**Questions:**

1) Why do spherical coordinates lead to instabilities? Could not follow the explanation in footnote 1, specifically the point about ordering of the angles and under certain circumstances subsequent angles having no influence on the orientation. Could the authors clarify this point? A visualization would be greatly beneficial in this regard.

2) Lines (242-243) : “On the Pascal VOC dataset, we do not compare ARI-BG scores, as we have found that a simple baseline significantly surpasses previously reported .….” . I didn’t understand the reasoning behind this choice.

3) (related to weakness #2) Why can’t the proposed method segregate 2 objects of the same color but spatially well separated? If the network has learned a simple rule using just a single feature like color or shape to perform binding, that would be extremely limiting in the general case.

4) (related to weakness #2) Could the authors show the grouping performance (FG-ARI, FULL-ARI) for the colorized ‘Shapes’ dataset without (simply using RGB images as inputs) the use of depth channel information? Have the authors performed an ablation to quantify how much the depth channel information assists the network to predict the object identities?

5) How does the proposed method work on colored multi-object datasets like Tetrominoes, Multi-dSprites or CLEVR that are the first benchmark suite typically used for evaluation in the object-centric literature?

6) Table 1 (Pascal VOC dataset results), compared to the baseline models which use pre-trained DINO features in the encoder module (i.e. DINOSAUR Transformer/MLP) how much instance-level grouping has been achieved through the use of RF? It’s known that the DINO features already possess a high-level of specialization to semantic classes and therefore can perform semantic grouping (by inspecting the attention masks after conditioning on the CLS token). In this regard, SlotAttention and SLATE baselines cannot be considered like-for-like since they still use the pixel reconstruction objective and are trained from scratch. Short point being that how much instance-level grouping is being learned through the use of RF on top of the semantic-level grouping that is  already captured by the pretrained DINO features?

**Limitations:**

Authors discuss limitations of their method, in particular that the capacity to represent multiple objects is currently limited to the output space of the autoencoder and that the methods still lag behind slot-based methods such as DINOSAUR.


References:

[1] “Complex-Valued Autoencoders for Object Discovery”, Lowe et. al, TMLR 2022.

---

> ### Author Rebuttal · Authors · 2023-08-09
>
> Thank you for your constructive review. We would like to take the opportunity to respond to the questions and concerns that you have posed.
>
> ### Weaknesses
>
> **1) How novel is this evaluation procedure compared to the one used by CAE?**
>
> While our proposed evaluation method closely resembles the CAE's evaluation method mathematically - merely requiring the addition of a weighted sum with binary weights - its conceptual significance cannot be understated: it avoids the trivial solutions described in lines 157-160 of the paper that would otherwise make a fair assessment of our approach on multi-channel images impossible.
>
> **2) Depth masks provide a strong supervision signal**
>
> Providing depth information is one possible way to prevent the Rotating Features from grouping together objects of the same color. Alternatively, we show that using features from a pretrained vision transformer works equally well. This approach, utilized by prior work for object discovery [1], is arguably more applicable in practice.
>
> **3) The experiment on FoodSeg103 does not meaningfully test the instance-level grouping ability.**
>
> Unfortunately, the FoodSeg103 dataset, like most semantic segmentation datasets, does not provide instance-level ground truth segmentation masks. This makes it challenging to meaningfully examine the instance-level grouping ability. Nonetheless, we believe that the class-level grouping performance of the Rotating Features on this dataset is noteworthy. For one, the DINO features exhibit a specialization towards the semantic class of a single object per scene when conditioned on the CLS token. In contrast, our experiments assess the performance in extracting multiple objects per scene, without feeding the CLS token into our network. Additionally, the DINO features are only used to set the input magnitudes, with input orientations being set to a fixed value. We evaluate the grouping learned by the output orientations, making our assessment independent of any grouping that may be extractable directly from the DINO features.
>
> ### Questions
>
> **1) Why do spherical coordinates lead to instabilities?**
>
> We will replace the example given in the paper with the following one: When a vector's magnitude is zero, the angular coordinates can take any value without changing the underlying vector. As our network applies ReLU activations on the magnitudes, this singularity may occur regularly, hindering the network from training effectively.
>
> **2) “On the Pascal VOC dataset, we do not compare ARI-BG scores”. I didn’t understand the reasoning behind this choice.**
>
> In Appendix D.2, we provide a more in-depth discussion on this.
>
> **3) Why can’t the proposed method segregate 2 objects of the same color but spatially well separated?**
>
> In its current form, Rotating Features do not include any inductive bias to enforce a spatial separation between object representations. We believe that this would be an interesting future direction to explore, as it could provide an alternative to our proposed solution that utilizes higher-level input features to perform binding on complex, real-world inputs.
>
> **4) Could the authors show the grouping performance for the colorized ‘Shapes’ dataset without depth information?**
>
> These results can be found in Figure 6, and Appendix Table 7 and Figure 14. They indicate that the network benefits strongly from the additional depth information.
>
> **5) How does the proposed method work on colored multi-object datasets typically used in the object-centric literature?**
>
> In response to the reviewer's suggestion, we evaluate the performance of the proposed Rotating Features model on the Multi-dSprites and CLEVR datasets, comparing it to its predecessor — the CAE model [2]. The outcomes are presented in the table below. Note that since the original CAE was only applicable to grayscale images, we combine it with our proposed evaluation procedure to make it applicable to multi-channel input images, which we denote as CAE*.
>
> |Dataset|Model|ARI-BG|
> |--|--|--|
> |Multi-dSprites|CAE*|0.371$\pm$0.056|
> ||Rotating Features|0.888$\pm$0.015|
> |CLEVR|CAE*|0.289$\pm$0.042|
> ||Rotating Features|0.664$\pm$0.013|
>
> The results indicate that the Rotating Features model significantly surpasses the performance of the CAE*, demonstrating good object separation on these two datasets. However, the results still lack behind the state-of-the-art. The qualitative examples in Figure 3 of the PDF uploaded alongside the general response show that Rotating Features encounter the same issue here, as we have seen with the colored 4Shapes dataset: objects of the same color tend to be grouped together. As demonstrated with the RGB-D version of the colored 4Shapes dataset and with our results using pretrained DINO features on real-world images, this issue can be addressed by using higher-level input features.
>
> **6) How much instance-level grouping is being learned on top of the semantic-level grouping that is already captured by the pretrained DINO features?**
>
> As described above, Rotating Features do not directly make use of the semantic specialization of the DINO features that may be found when conditioning on the CLS token. Nonetheless, we will add another baseline from the DINOSAUR paper [1] that allows for a direct comparison between the grouping that may be inherent to the DINO features and the grouping achieved by the Rotating Features model. This baseline applies k-means directly to the DINO preprocessed features and achieves an MBO$_i$ score of 0.363 and MBO$_c = 0.405$. For comparison, Rotating Features achieve scores of MBO$_i = 0.407$ and MBO$_c = 0.460$. We therefore conclude that the Rotating Features improve over the instance-level and semantic-level grouping that may be inherently present within the pretrained DINO features.
>
> ---
> [1] Maximilian Seitzer, et al. Bridging the gap to real-world object-centric learning. ICLR, 2023.
>
> [2] Sindy Löwe, et al. Complex-valued autoencoders for object discovery. TMLR, 2022.

---

> > ### Comment · Reviewer_r4Rq · 2023-08-17
> > **Rebuttal Response**
> >
> > I thank the authors for their responses to my questions. I still believe the limitations of the paper should be more pronounced in the main text. In particular, the fact that the model relies on the depth masks and the differentiation of semantic vs instance segmentation. In the figures provided in the paper there is not a single case of a figure containing multiple instances of the same class such that the method separates them. This should be made more obvious in the text. I'm happy to stick to my current rating of the paper.

---

### Official Review · Reviewer_e8db · 2023-07-07

**Soundness:** 3 good
**Presentation:** 3 good
**Contribution:** 4 excellent
**Rating:** 8
**Confidence:** 2

**Summary:**

The paper presents a new approach for extracting objects from distributed representations, based on a binding mechanism called 'rotating features' that extends previous phase-based binding notions to a much higher dimension binding space, and avoids the use of separate slots for individual objects, showing promising results and scaling behavior with both toy and natural-image data sets.

**Strengths:**

The exploration of alternatives to slot-based schemes, so prevalent in transformers and related contemporary architectures, for addressing how neural networks can successfully extract distinct, coherent representations of objects and other entities is an important contribution in itself.  For brain science, it is important because the brain is unlikely to have explicit slots of the kind that can be built into artificial networks.  For AI, this step may be equally important, as it allows for the possibility of more graded approaches to objecthood that could be important for capturing the kind of graded objecthood of many aspects of the natural world, and of avoiding some of the potential brittleness and arbitrariness of imposing slot-based approaches to structured objects made up of sub-objects.

The model seems to produce fairly impressive results compared to an alternative much more complex transformer models and beats other comparison models as well.  The excellent training time, and the possibility of avoiding explicit k-means clustering for segmentation, and the availability of uncertainly maps all seem like desirable properties of the model.

**Weaknesses:**

I have chosen to ask questions rather than express statements about weaknesses because I found that there were important features of the model and of the comparisons that I simply could not fully understand without going to source papers on DINOSAUR and the binding mechanism.  I do consider it a weakness of the paper that I was not able to understand these things better, and my rating and confidence would be increased further if these questions were addressed in the rebuttal and revision of the paper.

**Questions:**

I found it a bit difficult to be sure I was seeing fair comparisons in table 1.  If I understand correctly, DINOSAUR MLP is much simpler than the Rotating Features CNN.  Is it slot based in some way?

I have found it difficult to understand how the binding mechanism described on p 4 works and I did not feel I got much out of figure 3.  perhaps the problem lies in my lack of understanding of the meaning of the superscripts on z_in.  Another source of confusion is the statement that the extra input dimensions of x (in line 134) are all set to 0, such that they don't seem to be capable of having any effect.  Why are these dimensions then needed?  I see that learned bias weights apply to the output of f_w and that there are R^(n x d_out) of these.  The references to Lowe et al and Reichert & Serre should not be my only source of an understanding of how the mechanism works.

**Limitations:**

The limitations as stated seem valid, and I look forward to seeing where attempts to address these limitations will lead.

---

> ### Author Rebuttal · Authors · 2023-08-09
>
> Thank you for the feedback from your thoughtful review. We would like to take this opportunity to address the two questions you posed:
>
> **I found it a bit difficult to be sure I was seeing fair comparisons in table 1. If I understand correctly, DINOSAUR MLP is much simpler than the Rotating Features CNN. Is it slot based in some way?**
>
> The DINOSAUR MLP model is a slot-based model, which combines Slot Attention with a spatial broadcast decoder. In essence, it encodes the DINO preprocessed features into slots, and subsequently decodes each slot individually via an MLP decoder. The predicted slot mask from this decoder is then used to recombine the individual reconstructions, and to evaluate the object discovery performance of this model. We will clarify our description in the paper to make the comparison between the models more comprehensible.
>
> **I have found it difficult to understand how the binding mechanism described on p.4 works, and I did not feel I got much out of figure 3. perhaps the problem lies in my lack of understanding of the meaning of the superscripts on $z_{in}$. Another source of confusion is the statement that the extra input dimensions of $\mathbf{x}$ (in line 134) are all set to 0, such that they don't seem to be capable of having any effect. Why are these dimensions then needed? I see that learned bias weights apply to the output of $f_{\mathbf{w}}$ and that there are $\mathbb{R}^{n \times d_{\text{out}}}$ of these.**
>
> The rotating feature vector $z_{in} \in \mathbb{R}^{n \times d_{\text{in}}}$ has $n$ rotating dimensions and $d_{\text{in}}$ feature dimensions, i.e., channels. Our data does not contain any rotating information, so we introduce the additional dimensions by padding the input with zeros. Your observation regarding the biases is correct - they serve as the model's mechanism that allows it to rotate the initial features, and thus to make use of the additional dimensions and to push them away from zero. Essentially, every feature dimension has the capacity to learn a distinct orientation offset through this bias.
>
> Without the binding mechanism, the model would fail to learn to leverage these additional dimensions, as they inherently do not have a strong effect on the computations. The binding mechanism ensures that features with similar orientations are processed together, while features with dissimilar orientations are essentially masked out. This allows the network to create separate streams of information that it can process separately - which naturally leads to the emergence of object-centric representations.
>
> Regarding Figure 3, we agree that the superscripts on $z_{in}$ are ambiguous. We will amend the figure caption, improving notation and description, to the following:
>
> Effect of the binding mechanism. We start by randomly sampling two column vectors $\mathbf{a}, \mathbf{b} \in \mathbb{R}^n$ with $\left\lVert{\mathbf{a}}\right\rVert_2 = \left\lVert{\mathbf{b}}\right\rVert_2 = 1$. Assuming $d_{\text{in}} = 3, d_{\text{out}} = 1$ and $f_{\mathbf{w}}$ is a linear layer, we set $z_{in} = \left[ \mathbf{a}, \mathbf{a}, \mathbf{b} \right]$, weights $\mathbf{w} = \left[ \frac{1}{3}, \frac{1}{3}, \frac{1}{3} \right]^T $and biases $b = \left[0, ..., 0\right]^T$. Then, we plot $m_{bind}$ and $\left\lVert{\psi}\right\rVert_2$ on the y-axis. These denote the magnitudes of the layer's output before the application of the activation function, with (blue) and without (orange) the binding mechanism. Without the binding mechanism, misaligned features are effectively subtracted from the aligned features, resulting in smaller output magnitudes. The binding mechanism masks out features with dissimilar orientations, reducing this effect and leading to consistently larger magnitudes in $m_{bind}$. In the most extreme scenario, features with opposite orientations (i.e., with a cosine similarity of -1) are cancelled out by the binding mechanism, as the output magnitude ($\frac{2}{3}$) would remain the same if $z_{in} = \left[ \mathbf{a}, \mathbf{a}, \mathbf{0} \right]$.

---

> > ### Comment · Reviewer_e8db · 2023-08-12
> > **Thanks for responses, new experiment strengthens findings further**
> >
> > I thank the authors for their responses to my questions.  The innovative nature of this work makes it exciting and challenging, and the new experiment demonstrating generalization to more objects further underscores the promise of the approach.  I certainly continue to believe this paper deserves the attention of the community.  While the results may not yet be ground-breaking, the approach and direction certainly are.

---

### Author Rebuttal · Authors · 2023-08-09

We thank the reviewers for their constructive feedback. We are delighted to see the reviewers recognize that the exploration of alternatives to slot-based schemes as studied in our paper is important (e8db, r4Rq) and interesting (cH5o), and that our work may stimulate many interesting directions for future work (cH5o). Further, the reviewers have noted that the experiments are well performed (r4Rq), showing that the proposed Rotating Features achieve promising results (e8db, cH5o, VTkG) while being very efficient to train (e8db, CH5o). Additionally, reviewer r4Rq stated that our paper is very well written.

We want to use this general response to highlight an additional experiment that we have conducted following suggestions of reviewers cH4o and VTkG. This experiment shows that Rotating Features can generalize beyond the number of objects observed during training, even when the number of objects in the test images is unknown.

### Generalization to more objects

We conduct an experiment to assess the flexibility of Rotating Features in segmenting varying numbers of objects, thereby testing its ability to generalize beyond the number of objects observed during training. Additionally, this experiment evaluates the adaptability of the model when the number of objects in a scene is unknown.

To train the Rotating Features model, we make use of a modified version of the 10Shapes dataset. This version includes the same ten unique shapes as the original. However, only six of these shapes are randomly selected to appear in each image.

Post-training, we test the trained model with a range of variants of this dataset, each displaying between four and ten objects per image. We present the results in Figure 1 in the PDF uploaded alongside this general response. First, we observe that when the number of objects is known and $k$ for $k$-means is set accordingly, the performance of the Rotating Features model is best when fewer objects are present in each image, and decreases as more objects are added. However, considering the increase in difficulty when incorporating more objects in a scene of a fixed size, the model maintains a relatively stable performance across various numbers of objects per scene. Second, when the number of objects in a scene is not known, and we apply $k$-means with a fixed value of $k=7$ (corresponding to the number of objects observed during training, plus one for the background), performance degrades considerably the more the true number of objects deviates from the fixed value of $k$. However, this problem can be circumvented by using agglomerative clustering, and by setting the distance threshold on the training dataset to reflect the best performance when there are six objects in a scene. Using the same threshold across all test settings maintains consistent performance for varying numbers of objects in each scene, albeit slightly inferior to the $k$-means baseline, with $k$ representing the true number of objects.

In summary, our results suggest that Rotating Features can generalize beyond the number of objects observed during training, even when the number of objects in the test images is not known. We will include this experiment in the revised paper.

---

> ### Comment · Reviewer_cH5o · 2023-08-16
> **Comment on additional experiment**
>
> Thanks to the authors for sharing these additional results. It is very good to see that the method can generalize well to a larger number of objects, even without knowing the number in advance. Is it possible to perform a comparable experiment with slot attention (or another slot-based architecture)? This would be very informative, as it would be good to know what the tradeoffs are for the two approaches with respect to flexibility and generalization.

---

> > ### Author Response · Authors · 2023-08-18
> >
> > Thank you for your response. Following your suggestion, we have conducted a comparable experiment to test the generalization performance of the Slot Attention model as proposed by [1]. For this, we follow the same training and testing procedure as before, i.e. we train the model on a variant of the 10Shapes dataset in which each image contains six objects, and subsequently test it on images containing between four and ten objects.
> >
> > The results are summarized in this table:
> > |  Number of Objects          | 4     | 5  | 6 | 7 | 8  |  9 | 10 |
> > |--|--|--|--|--|--|--|--|
> > | **Number of Objects known**  |  |  |  |  |  |   |  |
> > | Slot Attention,  \#slots = \#objects+1                    | 0.968 | 0.965 | 0.957 | 0.944 | 0.926 | 0.897 | 0.856 |
> > | Rotating Features with k-means, k=\#objects+1             | 0.957 | 0.944 | 0.927 | 0.909 | 0.889 | 0.867 | 0.843 |
> > | **Number of Objects unknown**  |  |  |  |  |  |   |  |
> > | Slot Attention, \#slots = 7               | 0.981 | 0.975 | 0.957 | 0.912 | 0.837 | 0.753 | 0.670  |
> > | Rotating Features with k-means, k=7          | 0.893 | 0.921 | 0.927 | 0.889 | 0.836 | 0.777 | 0.719 |
> > | Rotating Features with agglomerative clustering        | 0.936 | 0.919 | 0.901 | 0.879 | 0.856 | 0.829 | 0.798 |
> >
> > As we can see, when the number of objects is known, Slot Attention slightly outperforms the Rotating Features model for all tested numbers of objects. When the number of objects is unknown, however, and we set the number of slots in Slot Attention to the number of objects observed during training (plus one for the background), we see its performance deteriorate as the number of objects per image increases. When testing on ten objects per image, Slot Attention's performance drops to an ARI-BG score of 0.670, while the Rotating Features model with agglomerative clustering achieves a score of 0.798.
> >
> > ---
> > [1] Locatello, Francesco, et al. "Object-centric learning with slot attention." NeurIPS, 2020.

---

> > > ### Comment · Reviewer_cH5o · 2023-08-18
> > > **Comment on further experiments**
> > >
> > > Thanks to the authors for performing this last-minute experiment. I think these results will enrich the paper, and it's good to see that the method performs so well when the number of objects is unknown.

---

> > > ### Comment · Reviewer_cH5o · 2023-08-18
> > > **Comment on further experiments**
> > >
> > > Thanks to the authors for performing this last-minute experiment. I think these results will enrich the paper, and it's good to see that the method performs so well when the number of objects is unknown.

---

### Decision · Program_Chairs · 2023-09-21

**Decision:**

Accept (oral)

**Comment:**

The reviewers were unanimous that this paper should be accepted – with strong support from two of them. There was unanimous agreement that the work is significant and that the method presented is both novel and interesting as an alternative to the popular slot-based approaches. There was also agreement that all experiments are well conducted and the presented method compares favorably to the considered baselines. The AC agrees and recommends the paper to be accepted.